# Hierarchical Demonstration Order Optimization for Many-shot In-Context Learning

**Yinhan He**[*]
University of Virginia
Charlottesville, VA
nee7ne@virginia.edu

**Wendy Zheng**[*]
University of Virginia
Charlottesville, VA
ncd9cf@virginia.edu

**Song Wang**
University of Central Florida
Orlando, FL
song.wang@ucf.edu

**Zaiyi Zheng**
University of Virginia
Charlottesville, VA
sjc4fq@virginia.edu

**Yushun Dong**
Florida State University
Tallahassee, FL
yd24f@fsu.edu

**Yaochen Zhu**
University of Virginia
Charlottesville, VA
uqp4qh@virginia.edu

**Jundong Li**
University of Virginia
Charlottesville, VA
jl6qk@virginia.edu

## Abstract

In-Context Learning (ICL) is a technique where large language models (LLMs) leverage multiple demonstrations (i.e., examples) to perform tasks. With the recent expansion of LLM context windows, many-shot ICL (generally with more than 50 demonstrations) can lead to significant performance improvements on a variety of language tasks such as text classification and question answering. Nevertheless, ICL faces the issue of demonstration order instability (ICL-DOI), which means that performance varies significantly depending on the order of demonstrations. Moreover, ICL-DOI persists in many-shot ICL, validated by our thorough experimental investigation. Current strategies for handling ICL-DOI are not applicable to many-shot ICL due to two critical challenges: (1) Most existing methods assess demonstration order quality by first prompting the LLM, then using heuristic metrics based on the LLM's predictions. In the many-shot scenarios, these metrics without theoretical grounding become unreliable, where the LLMs struggle to effectively utilize information from long input contexts, making order distinctions less clear. (2) The requirement to examine all orders for the large number of demonstrations is computationally infeasible due to the super-exponential complexity of the order space in many-shot ICL. To tackle the first challenge, we design a demonstration order evaluation metric based on information theory for measuring order quality, which effectively quantifies the usable information gain of a given demonstration order. To address the second challenge, we propose a hierarchical demonstration order optimization method named `HIDO` that enables a more refined exploration of the order space, achieving high ICL performance without the need to evaluate all possible orders. Extensive experiments on multiple LLMs and real-world datasets demonstrate that our `HIDO` method consistently and efficiently outperforms other baselines. Our code project can be found at `https://github.com/YinhanHe123/HIDO/`.

---

[*]These authors contributed equally to this work.

39th Conference on Neural Information Processing Systems (NeurIPS 2025).

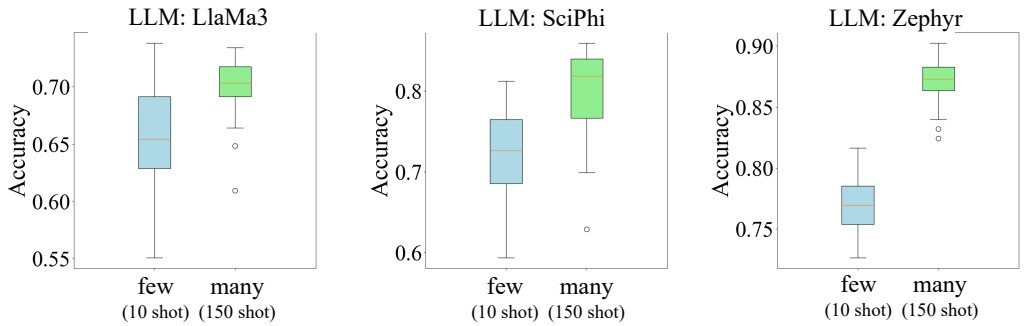

Figure 1: Accuracy of many-shot ICL (150 shots) in green and few-shot ICL (10 shots) in blue on dataset TREC. We randomly select demonstration orders and average the number of times the model predicts the correct answer for 256 queries. See more results in Appendix F.3.

# 1 Introduction

Large language models (LLMs) have demonstrated remarkable performance in few-shot In-Context Learning (ICL), where LLMs adapt to new tasks by incorporating demonstrations (examples) in the input prompt without additional fine-tuning [3, 8, 47]. Recent advances in research have vastly expanded the context windows of LLMs, paving the way for many-shot ICL [1, 12, 15, 2, 25]. This approach, typically involving more than 50 demonstrations, has achieved significant performance gains in various NLP tasks, including text classification [24] and question answering [16]. However, a critical challenge in few-shot ICL is *demonstration order instability* (ICL-DOI): the significant variance in ICL performance when the same set of demonstrations is arranged in different orders [21]. For instance, [21] claims that for a text classification task, different orders can cause performance to fluctuate dramatically, ranging from random guessing to 90% accuracy. Unfortunately, exploratory experiments (Fig. 1) show that the ICL-DOI phenomenon persists in many-shot ICL.

Several studies tackle the issue of ICL-DOI in few-shot scenarios. One thread of research design stabilization methods to reduce ICL performance variance w.r.t. different demonstration orders [4, 45, 41], while others search for the optimal demonstration order that maximizes ICL performance [21, 43, 19]. Although these proposed methods achieve satisfying performance under few-shot ICL, they cannot be easily adapted to many-shot scenarios [1] due to two fundamental challenges: (1) **Lack of precise quality-measuring metric:** Current studies [21, 43] assess order quality by examining LLM responses to sample queries with ordered demonstrations as context. These evaluations rely on heuristic properties like answer distribution diversity but suffer from subjective judgments. Additionally, LLMs exhibit primacy and recency bias, giving more attention to content at the beginning and end of large context windows [18]. This creates a challenge in many-shot scenarios where numerous demonstrations in the middle cause only subtle performance differences, making it difficult to compare different orders. (2) **Infeasibility of evaluating all demonstration orders:** Unlike few-shot ICL, it is infeasible to evaluate all demonstration orders in many-shot scenarios exhaustively. This is because, evaluating one demonstration order requires at least one LLM inference call, which is both costly and time-consuming. Furthermore, the demonstration order space expands super-exponentially ($n!$) with the increase of demonstration numbers.

In this paper, we address the issue of ICL-DOI in many-shot ICL by searching for an effective demonstration order. Specifically, to tackle the first challenge, we introduce In-Context Demonstration Order $V$-information (ICD-OVI). This metric, grounded in information theory, measures how effectively an LLM, with a certain ordered demonstration as context, extracts usable information from a query to infer its corresponding answer. To address the second challenge, we introduce a **HI**erarchical **D**emonstration **O**rder optimization (**HIDO**) framework that enables more refined exploration in the order space, thus achieving satisfactory performance without evaluating all possible orders.

We summarize our contributions as follows: (1) **Theoretically Justified Novel Metric**: We introduce a novel information-theoretic metric, ICD-OVI, which evaluates the quality of a demonstration order by the usable information extracted from a query, with the ordered demonstration as context, to infer its answer. (2) **Fundamental Optimization Framework**: Based on our ICD-OVI demonstration order quality metric, we propose an efficient hierarchical demonstration order optimization framework termed `HIDO` for in-context learning with vast demonstration permutation spaces. (3) **Extensive**

**Empirical Evaluations**: We conduct extensive experimental investigations on multiple LLMs and real-world datasets of various semantic scenarios, demonstrating the effectiveness and efficiency of our demonstration order optimization framework `HIDO`.

## 2 Preliminaries and Problem Definition

*Notations.* We denote a demonstration as $d := (q, a)$, where $q$ is its query, and $a$ is its answer. Throughout this work, we follow existing literature [21, 43] by considering the query $q$ as a multiple choice question and $a$ as a **single-token** answer (e.g., "A," "B" or "C") indicating the choice. For example, in a sentiment classification task, a demonstration $(q, a)$ might have query $q$ as "Please analyze the sentiment of this sentence: 'Jerry wins the game!' A. Positive; B. Negative" with corresponding answer $a$ as "A." To transform $(q, a)$ into text format for LLM input, we apply a transformation $\mathcal{T}$ that organizes the pair into a standard format: $\mathcal{T}(q, a) =$ "input: "$q$, "type: "$a$. Given that $T_i = \mathcal{T}(q_i, a_i)$ denotes the transformed text of the $i$th in-context demonstration, the demonstration set to be ordered is written as $\mathcal{D} := \{T_i\}_{i=1}^n$, where $n$ is the number of demonstrations. Each demonstration is assumed i.i.d. samples drawn from an underlying input data distribution $\mathcal{F}$ that we cannot directly access. We define $\pi \in S_n$ as an order permutation function ($S_n$ is the collection of all possible permutations), which maps the set $\{1, ..., n\}$ bijectively to itself. We use $\Pi(\mathcal{D}) := T_{\pi(1)} \oplus ... \oplus T_{\pi(n)}$ to represent concatenated demonstrations ordered by permutation $\pi$, where $\oplus$ represents text concatenation. Finally, we denote $\mathcal{P}_{\text{LLM}}(\cdot|q)$ as the LLM prediction (i.e., logits vector) when we prompt the LLM with query $q$.

*Preliminaries.* We first introduce the existing ICL demonstration order quality metrics, then provide the theoretical background for our proposed method. All existing metrics measure the demonstration order quality with an important tool: a set of *probing samples* $\hat{\mathcal{D}} := \{(\hat{q}_i, \hat{a}_i)\}_{i=1}^m$ (where $m$ is the number of probing samples) generated by the same LLM that we perform ICL on. These probing samples mimic samples drawn from the inaccessible distribution $\mathcal{F}$, thus providing insight into $\mathcal{F}$.

The probing samples are generated in two steps: (1) For a given order $\pi$ of demonstrations, we concatenate the demonstrations into $\Pi(\mathcal{D})$ and prompt the LLM with this sequence to generate new samples that are similar to the demonstrations, which are called probing samples. (2) We repeat this process for all possible permutations of demonstration orders ($\pi \in S_n$). The complete set of probing samples $\hat{\mathcal{D}}$ is then the collection of all samples generated across all permutations.

**Existing Method 1: GlobalE.** Given ordered demonstrations $\Pi(\mathcal{D})$, GlobalE first calculates the frequency distribution of the LLM's predicted answers for queries in the probing samples with $\Pi(\mathcal{D})$ as context: $\mathbf{f} = \frac{1}{m} \sum_i \mathbb{I}[\arg\max \mathcal{P}_{\text{LLM}}(\cdot|\Pi(\mathcal{D}) \oplus \hat{q}_i)]$. Here, $\mathbb{I}[\cdot]$ is the indicator function that transforms an integer to its corresponding one-hot vector with length equal to the number of possible answers. GlobalE measures the diversity of answers by the distribution entropy $H_{X \sim \mathbf{f}}(X)$. Lu et.al [21] claim that answer diversity maintains a high positive correlation with the accuracy of LLM predictions empirically. Therefore, demonstrations with higher GlobalE values are considered preferable.

**Existing Method 2: LocalE.** LocalE measures the average conditional entropy of an LLM's predictions (i.e., the logits vectors) when prompted with the ordered demonstrations and probing queries. It is calculated as $\frac{1}{m} \sum_i H_{X \sim \mathcal{P}_{\text{LLM}}}(X|\Pi(\mathcal{D}), \hat{q}_i)$. Unlike GlobalE, which examines answer frequency distribution across all probing samples, LocalE focuses on the uncertainty in the model's predictions for individual samples. A higher LocalE value indicates that the LLM is less confident in its predictions, which helps prevent overconfidence and poor calibration. However, GlobalE and LocalE are heuristic metrics derived from empirical observations. They also do not utilize the answers from probing samples since they cannot verify the correctness of these answers.

**Existing Method 3: PDO.** Another metric is the probability distribution optimization (PDO) metric [43]. This metric is designed based on the assumption that well-ordered in-context examples should produce answer frequency distributions that match a prior distribution, which is believed to approximate the actual answer distribution. PDO measures demonstration quality by calculating the discrepancy between the answer frequency distribution produced by the LLM and the prior distribution, formally written as $\text{KL}\left(\frac{1}{m} \sum_i \mathcal{P}_{\text{LLM}}(\cdot|\Pi(\mathcal{D}) \oplus \hat{q}_i) \| U_{\mathcal{A}}\right)$. In this formula, $\text{KL}(\cdot\|\cdot)$ represents the KL divergence, and $U_{\mathcal{A}}$ is the human-determined prior probability distribution (typically a uniform distribution) over the answer space $\mathcal{A}$. However, this approach has limitations because

the prior distribution of probing samples' answers may not accurately reflect the actual input data distribution. This discrepancy has led to debates about the effectiveness of the PDO metric.

**Theoretical Foundation: $\mathcal{V}-$usable information.** Our design of ICL demonstration order metric is inspired by information theory, specifically $\mathcal{V}$-usable information [42, 17], a widely recognized information-theoretic metric measuring the amount of information an ML model $f$ can capture from input queries random variable (r.v.) $Q$ to predict their corresponding answers r.v. $A$. Specifically, for a predictive family $\mathcal{V}$ (i.e., the set of a model's all possible parameter configurations), the $\mathcal{V}$-usable information is defined as $H_{\mathcal{V}}(A|\emptyset) - H_{\mathcal{V}}(A|Q)$, where

$$
\begin{aligned}
H_{\mathcal{V}}(A|Q) &= \inf_{f \in \mathcal{V}} \mathbb{E}_{(q,a)\sim\mathcal{F}}[-\log_2 f(\cdot|q)], \\
H_{\mathcal{V}}(A|\emptyset) &= \inf_{f \in \mathcal{V}} \mathbb{E}_{(q,a)\sim\mathcal{F}}[-\log_2 f(\cdot|\emptyset)].
\end{aligned}
\tag{1}
$$

Here, $\mathcal{F}$ is the input data distribution, $f(\cdot|q)$ is the predicted answer distribution given query $q$, and $f(\cdot|\emptyset)$ represents the model's prediction without seeing the query (using only prior knowledge). $H_{\mathcal{V}}(A|Q)$ and $H_{\mathcal{V}}(A|\emptyset)$ measures the expected log-loss of the optimal predictor in the family $\mathcal{V}$ when it has access to the query and when it does not have access to the query, respectively. Therefore, this metric, $H_{\mathcal{V}}(A|\emptyset) - H_{\mathcal{V}}(A|Q)$, quantifies how much additional information about the answer the model can extract when it sees the query compared to when it does not. This metric has multiple advantages: (1) *Interpretable*: It measures information amount (in units of "bits") that a model with predictive family $\mathcal{V}$ can capture from $Q$ to predict $A$, which is easily human-comprehensible. (2) *Computationally Viable*: Although the data distribution $\mathcal{F}$ is not accessible, the $\mathcal{V}-$usable information can be efficiently approximated by Monte Carlo with a theoretical precision guarantee [42]. (3) *Empirically Effective*: the metric is empirically proven with a high correlation with the correctness of the predicted answer [17, 44, 38]. Note that the $H_{\mathcal{V}}(\cdot)$ is similar in format compared with entropy $H(\cdot)$, however, existing work proves that using entropy $H(\cdot)$ will lose the advantage (2) and (3) for the $\mathcal{V}$-usable information [42, 17, 44, 38].

***Problem Definition***. Here, we formulate the in-context learning demonstration order optimization task as finding the order that minimizes the distribution discrepancy between the LLM output and the original input. Specifically, we have the following definition:

**Definition 1.** *For a demonstration data distribution $\mathcal{F}$, where each data sample is in the shape of (query, answer), given $n$ demonstrations i.i.d. drawn from $\mathcal{F}$, denoted as $\mathcal{D}$, we aim to find the demonstration order $\hat{\pi}$ of the $n$ demonstrations such that the answer prediction distribution produced by LLM approximates $\mathcal{F}$ (measured by "KL divergence [5]"), i.e.,*

$$
\hat{\pi} = \min_{\Pi} KL(\mathcal{P}_{LLM}(\cdot|\Pi(\mathcal{D}) \oplus q)||\mathcal{F}(\cdot|q)).
\tag{2}
$$

## 3 In-Context Demonstration Order $\mathcal{V}$-usable Information

Before introducing our proposed `HIDO` framework, we first present a novel evaluation metric called **In-Context Demonstration Order $\mathcal{V}$-Usable Information (ICD-OVI)**. Unlike existing heuristic metrics such as GlobalE [21], LocalE [21], and PDO [43], our ICD-OVI is grounded in information theory and measures how effectively an LLM extracts useful information from queries to predict their answers when given a specific demonstration order. Previous metrics rely primarily on empirical observations and often lack theoretical justification, making them less reliable for many-shot ICL scenarios. In these scenarios, similar demonstration orders produce only subtle differences in LLM performance, resulting in very similar metric values that make it difficult to distinguish quality between different orders. Our approach addresses this limitation by providing a information-theoretic metric that quantifies exactly how much a demonstration order helps the LLM learn from inputs.

### 3.1 Intuition and Definition

The key intuition behind ICD-OVI is simple: **the optimal demonstration order should help the LLMs extract maximum information from queries to produce correct answers**. Our metric quantifies this information extraction capability. ICD-OVI measures the usable information that an LLM can capture from ordered demonstrations $\Pi(\mathcal{D})$. First, we follow the theory of $\mathcal{V}-$usable information and define the predictive family corresponding to the ordered demonstrations as

$$
\mathcal{V}_{\Pi} := \{\mathcal{P}_{\text{LLM}}(\cdot|\Pi(\mathcal{D}) \oplus q)|q \in \mathcal{Q}_{\mathcal{F}}\} \cup \{\mathcal{P}_{\text{LLM}}(\cdot|q)|q \in \mathcal{Q}_{\mathcal{F}}\}.
\tag{3}
$$

This represents all possible LLM configurations achievable with the given ordered demonstrations as context. Here, $\mathcal{Q}_{\mathcal{F}}$ is the set of all possible queries from the data distribution, and the second term of Equ 3 is included to ensure $\mathcal{V}_\Pi$ satisfies the technical requirements of a valid predictive family [42]. With the predictive family $\mathcal{V}_\Pi$, we follow the theory of $\mathcal{V}-$usable information to derive the in-context demonstration order $\mathcal{V}-$usable information, i.e., ICD-OVI, as (see full derivation in Appendix D):

$$\text{ICD-OVI} = \mathbb{E}_{(q,a)\sim\mathcal{F}}[\log_2 \mathcal{P}_{\text{LLM}}(a|\Pi(\mathcal{D}) \oplus q) - \log_2 \mathcal{P}_{\text{LLM}}(a|\Pi(\mathcal{A}) \oplus \emptyset)], \qquad (4)$$

where $\Pi(\mathcal{A})$ is the concatenation of all answers in the order of $\pi$. Equ. 4 captures the difference between the model's ability (1) to predict answers when seeing both the ordered demonstrations and the query (first term); and (2) to predict answers without seeing any queries (second term). Essentially, this equation quantifies the expected information the LLMs can capture from a query for prediction with the ordered demonstrations as context.

## 3.2 Practical Implementation

Since we cannot directly access the data distribution $\mathcal{F}$, we approximate ICD-OVI using probing samples $\hat{\mathcal{D}}$ generated by the LLM. They effectively mimic samples that are drawn from $\mathcal{F}$. Formally,

$$\text{ICD-OVI} \approx \frac{1}{|\hat{\mathcal{D}}|} \sum_{i=1}^{m}[\log_2 \mathcal{P}_{\text{LLM}}(\hat{a}_i|\Pi(\mathcal{D}) \oplus \hat{q}_i) - \log_2 \mathcal{P}_{\text{LLM}}(\hat{a}_i|\Pi(\mathcal{A}) \oplus \emptyset)]. \qquad (5)$$

However, this approximation involves LLM-generated labels ($\hat{a}$) from probing samples, which may be incorrect and introduce bias. To address this, we use a technique from $\mathcal{V}$-usable information theory called the **point-wise $\mathcal{V}$-usable information threshold (PVI threshold, denoted as $\tau$)**.

Denoting $\text{PVI}_{(\hat{q},\hat{a})}$ as the probability subtraction in Equ. 5 for a single probing sample, the PVI threshold helps us determine if the label $\hat{a}$ of a probing sample $(\hat{q}, \hat{a})$ is reliable: (1) If $\text{PVI}_{(\hat{q},\hat{a})} \geq \tau$, the label $\hat{a}$ is highly likely correct; (2) If $\text{PVI}_{(\hat{q},\hat{a})} < \tau$, the label $\hat{a}$ is possibly incorrect. Prior work [9, 20] show empirically that this threshold effectively separates correct from incorrect labels across various datasets and LLMs. Hence, when calculating ICD-OVI with Equ. 5, we keep the original term for probing samples with reliable labels $\hat{a}$. For unreliable labels, we replace the term with its expectation over all possible labels to avoid relying on a single, potentially incorrect answer. If we call the above computation procedure for a probing sample as *point-wise* ICD-OVI (PICD-OVI), our ICD-OVI is the average of PICD-OVIs across all probing samples.

## 3.3 Theoretical Properties

The ICD-OVI enjoys properties summarized in the following theorem (see proof in Appendix E)

**Theorem 1.** *Given an LLM and demonstrations $\mathcal{D}$ drawn from data distribution $\mathcal{F}$, for any two ordered demonstrations $\Pi_1(\mathcal{D})$ and $\Pi_2(\mathcal{D})$, under mild conditions, if ICD-OVI($\Pi_1(\mathcal{D})$) > ICD-OVI($\Pi_2(\mathcal{D})$), then the LLM achieves better performance with $\Pi_1(\mathcal{D})$ as context than with $\Pi_2(\mathcal{D})$ when predicting answers for queries drawn from $\mathcal{F}$.*

This theorem establishes that our metric correctly identifies better-performing demonstration orders, providing theoretical justification for its use in optimizing ICL performance. Our ICD-OVI is the first information-theoretic metric for evaluating ICL demonstration orders, and inherits the beneficial properties of $\mathcal{V}$-usable information theory. We conduct extensive experiments in Section 5 to demonstrate the effectiveness of the ICD-OVI metrics.

***Remark.*** Although each probing sample $(\hat{q}, \hat{a})$ seems to require two separate LLM inference calls (one for $\Pi(\mathcal{D})\oplus\hat{q}$ and another for $\Pi(\mathcal{A})\oplus\emptyset$), we can optimize this process. For a given demonstration order, the term $\mathcal{P}_{\text{LLM}}(\hat{a}|\Pi(\mathcal{A}) \oplus \emptyset)$ only needs to be computed once and can be reused across all probing samples. This optimization ensures that ICD-OVI's computational complexity remains comparable to traditional heuristic metrics despite its theoretical advantages.

## 4 Methodology

Our HIDO, shown in Fig 2, first clusters the embeddings of the input demonstration texts and then performs $k$ iterations of hierarchical order optimizations. In each iteration, the process first determines

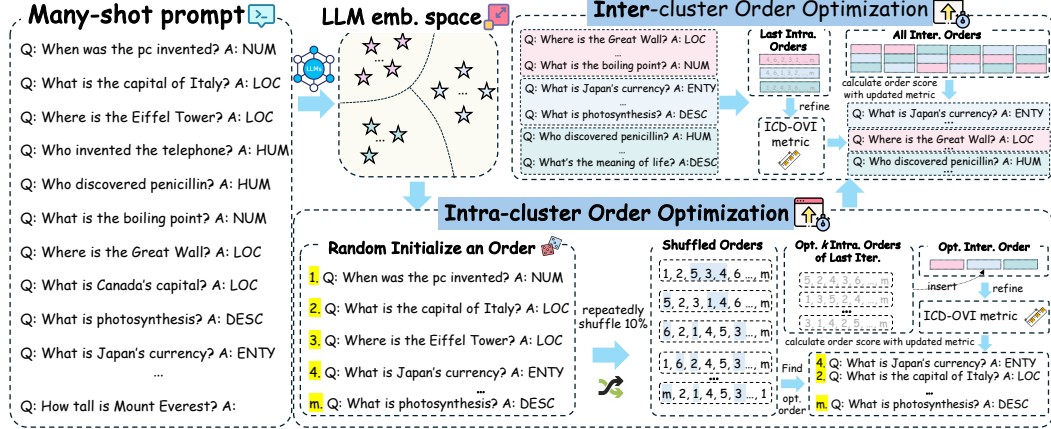

Figure 2: Overview of our proposed HIDO framework.

the near-optimal order within each demonstration cluster. Then, while maintaining these intra-cluster orders, it searches for the most effective order of the clusters themselves. This alternating focus on intra- and inter-cluster optimization may be iterated multiple rounds during which the probing samples are imporved (see detailed rationale in Section 4.4) to achieve more accurate assessment of the demonstration order quality using ICD-OVI.

## 4.1 Demonstration Clustering

Clustering demonstrations dramatically reduce the permutation space, allowing for more efficient search of the optimal order. Additionally, as the demonstrations within the same cluster are approximate in embedding space, varying demonstration orders cause less ICL performance variance. Thus, we apply a $K$-means algorithm [23] to the text embeddings of the demonstrations. These text embeddings are generated using the text embeddings API from [28]. We limit the number of clusters to be small (typically no more than four), as a larger number would cause a combinatorial explosion during HIDO's inter-cluster order optimization stage, where all possible orders are evaluated.

## 4.2 Intra-cluster Order Optimization

In Section 4.1, we restrict the cluster number to be small, which implies that demonstrations within one cluster, despite sharing similar LLM embeddings, can be large in quantity (e.g., 30 intra-cluster samples). Nevertheless, the intra-cluster demonstrations share proximate embeddings, which significantly decreases ICL performance variance when demonstration orders vary. This allows a less thorough order search while still achieving satisfactory precision. Hence, we design an intra-cluster demonstration order exploration strategy as follows (see the lower right hand side of Fig. 2): we first randomly generate a demonstration order, then in each iteration, we explore its "neighborhood" by randomly flipping 10% of its positions, which ensures variation between selected orders while constraining exploration within a radius of the original order, as measured by rank correlations [13].

**Theorem 2.** *Randomly flipping $K$ entries from a sequence of length $N$ keeps the rank correlation within a range characterized by the lower bound $1 - 6\sum_{i=1}^{K}(a_i - a_{K+1-i})^2/N(N^2 - 1)$ and upper bound $1$. The upper bound is achieved with a extremely low probability of $1/K!$ when the perturbed sequence is identical to the original sequence. (see proof in Appendix E)*

We evaluate each candidate intra-cluster order's quality using the ICD-OVI metric with a probing set generated by the examined LLM. We generate the probing samples with input demonstrations as context in three steps: We start with the top $k$ (a predefined hyperparameter) effective intra-cluster orders for the target cluster from the previous optimization iteration. For each of these $k$ intra-cluster order, we expand them to $k$ distinct orders for the entire set of the demonstrations by combining: *(i)* The optimal *inter-cluster order* from the previous iteration (fixed); *(ii)* The optimal intra-cluster orders for *all other clusters* from the previous iteration (fixed); *(iii)* The current candidate intra-cluster order for *the target cluster* being optimized. The $k$ distinct ordered demonstrations differ only in the order of demonstrations within the target cluster. We prompt the LLM with each ordered demonstrations to generate $k$ different probing sample sets. All probing samples collectively serve as the probing sample

Table 1: Metadata of the LLMs tested. "Lan. models", "Con. window" indicates the language models, and context window size.

| Lan. models | GPT-3.5T | GPT4oM | SciPhi | Zephyr | LlaMa3 |
|---|---|---|---|---|---|
| Con. window | 16,385 | 128,000 | 32,768 | 32,768 | 1,048,576 |
| Max output | 4,096 | 16,384 | n/a | n/a | n/a |
| Model size | 175B | n/a | 7B | 7B | 8B |

set to evaluate the candidate intra-cluster order using the ICD-OVI metric. By using probing samples derived from previous top-performing orders, we achieve more robust candidate order evaluation.

## 4.3 Inter-cluster Order Optimization

Having obtained the near-optimal demonstration orders within each cluster, we now find the optimal order of the clusters themselves. As we have limited the number of clusters (typically no more than four), it becomes feasible to evaluate all possible cluster orders, as illustrated in upper right hand side of the Fig. 2. Similar to the intra-cluster optimization process, we generate a probing set to evaluate each possible inter-cluster order with ICD-OVI metric. However, in this case, we employ all possible inter-cluster orders, while fixing the optimal intra-cluster demonstration orders obtained from the previous iteration. Specifically, we first consider all possible permutations of cluster orders, then prompt the LLM with this complete set of ordered demonstrations (combining the inter-cluster order being evaluated and the fixed optimal intra-cluster orders) to generate a probing set. This approach allows us to comprehensively assess different cluster arrangements while leveraging the optimized intra-cluster orders, potentially leading to a globally optimized demonstration order.

## 4.4 Dynamic Update of the Score Function

We perform multiple rounds of intra- and inter-cluster optimization, during which the *score function (ICD-OVI) is refined through updated probing sets* (see the upper and lower right hand side of Fig. 2). A higher-quality probing set reduces distribution discrepancy between probing and input data samples, enabling more precise ICD-OVI estimation. This further improves accuracy in identifying effective demonstration orders for answer prediction.

The procedure is separately introduced in the Section 4.2 and Section 4.3, therefore, we briefly conclude it as follows. In each iteration of in-context demonstration order optimization, we cache the top $k$ intra-cluster demonstration orders for all clusters. For intra-cluster optimization in the subsequent iteration, we apply the cached top $k$ intra-cluster demonstration orders for the cluster being optimized, while maintaining the optimal intra-cluster orders from the previous iteration for all other clusters. We combine these with the optimal inter-cluster order from the previous iteration to generate new probing sets. For inter-cluster optimization, we consider all possible cluster arrangements. For each arrangement, we apply the optimal intra-cluster demonstration orders obtained from the previous iteration to generate probing sets for evaluating each inter-cluster arrangement. By iteratively refining our probing sets for both intra-cluster and inter-cluster optimizations, we improve the evaluation accuracy progressively, leading to optimized orders over time.

## 5 Experiments

We first introduce our experimental setup. Then, we answer the following research questions about our proposed hierachical demonstration order optimization method HIDO via extensive experiments: **RQ1:** How does HIDO perform compared to existing demonstration order optimization methods across different datasets and language models? **RQ2:** What is the impact of each key component in HIDO on its overall performance? **RQ3:** How sensitive is HIDO to its main hyperparameters such as the number of clusters and the maximum number of optimization iterations?

### 5.1 Experiment Setup

Here, we introduce the various settings for our experimental evaluation.

Table 2: Performance of HIDO vs. baselines across datasets (best results in bold).

| | | AGNews | CB | CR | DBPedia | MPQA | MR | RTE | SST-5 | TREC |
|---|---|---|---|---|---|---|---|---|---|---|
| GPT-3.5T | GlobalE | 87.24 ± 0.60 | 46.43 ± 9.28 | 93.36 ± 0.39 | 95.70 ± 1.41 | 90.76 ± 0.81 | 93.62 ± 0.98 | **81.90** ± 0.90 | 54.56 ± 2.83 | 77.47 ± 6.56 |
| | LocalE | 89.06 ± 0.39 | 46.43 ± 9.28 | 93.10 ± 0.81 | 95.83 ± 1.37 | 89.97 ± 0.60 | 93.49 ± 1.19 | 80.86 ± 0.39 | 52.60 ± 4.77 | 78.65 ± 6.72 |
| | PDO | 89.32 ± 0.45 | 48.21 ± 7.78 | 93.23 ± 0.60 | 96.22 ± 1.80 | 89.97 ± 0.98 | 93.62 ± 0.98 | 80.60 ± 0.45 | 53.65 ± 3.52 | 76.69 ± 6.08 |
| | HIDO | **89.45** ± 0.39 | **51.19** ± 2.73 | **94.27** ± 0.23 | **97.92** ± 0.23 | **91.02** ± 0.78 | **94.27** ± 0.45 | **81.90** ± 0.60 | **54.95** ± 1.48 | **82.29** ± 1.63 |
| GPT-4oM | GlobalE | 83.07 ± 3.37 | 55.95 ± 1.03 | 93.36 ± 0.39 | 92.19 ± 2.17 | **87.50** ± 1.79 | 92.71 ± 0.45 | 85.16 ± 0.78 | 53.39 ± 2.48 | 83.33 ± 2.15 |
| | LocalE | 84.77 ± 0.78 | 55.95 ± 1.03 | **93.36** ± 0.68 | 92.19 ± 2.56 | 86.33 ± 3.73 | 92.32 ± 2.22 | 85.55 ± 1.35 | 53.26 ± 2.60 | 84.11 ± 1.76 |
| | PDO | 85.03 ± 2.00 | 55.36 ± 0.00 | 92.84 ± 0.60 | 92.19 ± 2.34 | 81.64 ± 1.41 | 92.84 ± 1.13 | 85.42 ± 1.63 | 52.86 ± 2.22 | 84.51 ± 2.60 |
| | HIDO | **85.81** ± 2.22 | **56.55** ± 1.03 | **93.36** ± 0.68 | 92.84 ± 0.81 | 86.85 ± 1.13 | **93.23** ± 0.60 | **86.33** ± 0.68 | **56.64** ± 3.20 | **86.59** ± 1.37 |
| SciPhi | GlobalE | 85.29 ± 0.81 | **92.26** ± 1.03 | 91.67 ± 0.60 | 96.09 ± 1.41 | 83.59 ± 0.68 | 93.88 ± 0.45 | 83.72 ± 2.39 | 54.69 ± 2.38 | 76.17 ± 7.32 |
| | LocalE | 86.59 ± 0.23 | **92.26** ± 1.03 | 92.32 ± 1.13 | 96.22 ± 0.81 | 85.16 ± 0.68 | 93.88 ± 0.23 | 83.98 ± 0.39 | 55.08 ± 1.70 | 76.69 ± 3.16 |
| | PDO | 86.07 ± 0.60 | **92.26** ± 1.03 | 91.02 ± 1.70 | 96.09 ± 1.41 | 84.77 ± 0.39 | 94.01 ± 0.23 | 83.72 ± 2.39 | 54.69 ± 2.38 | 76.17 ± 7.32 |
| | HIDO | **86.98** ± 0.45 | 90.48 ± 1.03 | **92.71** ± 0.60 | **96.88** ± 0.68 | **87.50** ± 0.78 | **94.27** ± 0.45 | **85.94** ± 0.78 | **57.16** ± 1.85 | **80.47** ± 0.78 |
| Zephyr | GlobalE | **89.71** ± 0.98 | 77.38 ± 5.15 | 93.23 ± 0.90 | 94.66 ± 2.00 | 86.07 ± 1.26 | 94.40 ± 0.45 | 82.16 ± 1.13 | 50.00 ± 0.68 | 84.38 ± 1.17 |
| | LocalE | 88.15 ± 0.23 | 73.21 ± 4.72 | 93.10 ± 1.13 | 96.22 ± 1.97 | 86.98 ± 0.60 | 94.66 ± 0.23 | 82.55 ± 0.81 | 48.18 ± 1.85 | 81.90 ± 4.30 |
| | PDO | 88.80 ± 0.81 | 77.38 ± 5.15 | 93.23 ± 0.90 | 94.66 ± 2.00 | 86.07 ± 0.60 | 93.10 ± 0.98 | 81.51 ± 1.93 | 50.00 ± 0.68 | 84.38 ± 1.17 |
| | HIDO | 89.32 ± 0.90 | **78.57** ± 1.79 | **94.01** ± 0.45 | **97.27** ± 0.68 | **87.76** ± 0.98 | **94.79** ± 0.60 | **82.55** ± 1.37 | **50.78** ± 2.07 | **86.46** ± 1.48 |
| LlaMa3 | GlobalE | 80.34 ± 4.95 | **94.64** ± 1.79 | 85.94 ± 3.20 | 93.49 ± 1.48 | 58.20 ± 2.34 | 92.84 ± 0.81 | 82.42 ± 0.39 | 39.19 ± 1.93 | 72.92 ± 1.48 |
| | LocalE | 83.72 ± 4.49 | 91.67 ± 2.06 | 85.68 ± 4.77 | 93.23 ± 0.23 | 54.82 ± 1.26 | 90.49 ± 0.81 | 82.81 ± 1.03 | 40.36 ± 4.21 | 73.18 ± 6.79 |
| | PDO | 77.73 ± 1.03 | **94.64** ± 1.79 | 85.16 ± 2.38 | 93.49 ± 1.48 | 52.21 ± 1.26 | 91.02 ± 1.35 | 82.94 ± 0.23 | 39.19 ± 1.93 | 72.92 ± 1.48 |
| | HIDO | **86.20** ± 2.29 | **94.64** ± 3.09 | **87.24** ± 2.39 | **94.27** ± 1.58 | **63.80** ± 7.64 | **93.49** ± 0.98 | **83.07** ± 0.98 | **40.62** ± 3.58 | **77.34** ± 3.73 |

***Baselines***: (1) **GlobalE**: Randomly select 24 orders and measure the entropy of the frequency distribution of the prediction labels on probing datasets [21]; (2) **LocalE**: Analogously to [21], randomly select 24 demonstration orders and calculate the average entropy of their predicted logits given by LLM. (3) **Probability Distribution Ordering (PDO)** [43]: Calculates the score function of a demonstration order as the KL divergence between the frequency distribution of the prediction labels generated by the LLM on probing datasets and the uniform distribution as the prior distribution.

***Datasets***: We adopt nine text classification datasets: AG's News Corpus (AGNews) [46], CommitmentBank (CB) [7], Customer Review (CR) [11], DBPedia Ontology Classification (DBPedia) [46], Multi-Perspective Question Answering (MPQA) [39], Movie Review (MR) [30], Recognizing Textual Entailment (RTE) [6], Stanford Sentiment Treebank-5 (SST-5) [34], and Text REtrieval Conference Question Classification (TREC) [37]. They cover various semantic scenarios, including sentiment classification and textual entailment. We sub-sample 256 instances from each dataset due to budget constraints.

***Large Language Models***: We adopt "GPT-3.5-Turbo-0125" [26] and "GPT-4o-Mini-2024-07-19" [27] from OpenAI, "SciPhi-Mistral-7B-32k" [33], "Zephyr-7b-beta" [36] and "LlaMa-3-8B-Instuct-Gradient-1048k" [32] from HuggingFace. We select those OpenAI models due to their affordability and the HuggingFace models due to their large context windows.

## 5.2 Effectiveness of HIDO

In this section, we aim to answer **RQ1**. In Table 2, we measure the accuracy of the output demonstration orders produced by HIDO and the baselines on various datasets and LLMs. We observe that HIDO achieves the highest prediction accuracy in most settings, proving the effectiveness of our framework. Notably, our method can achieve significant performance leads in GPT-3.5T on CB (51.19%), GPT-4oM on SST-5 (56.64%), SciPhi on TREC (80.47%), and LlaMa3 on MPQA (63.80%). Additionally, we make the following observations from Table 2: (1) **Model-agnostic**: HIDO achieves the best performance on both large and small LLMs, implying that our framework is model agnostic; it can be used on different models and find relatively high-performing orders.

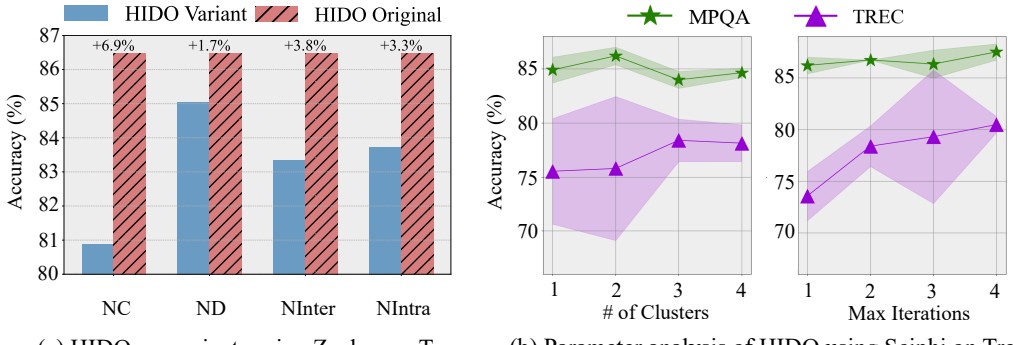

(a) HIDO vs. variants using Zephyr on Trec.  (b) Parameter analysis of HIDO using Sciphi on Trec.

Figure 3: Combined results of ablation study and parameter analysis.

(2) **Low variance**: In general, HIDO has a smaller variation in performance on most dataset model combinations in contrast to that of the baselines, especially in GPT-3.5T on CB (2.73%), GPT-4oM on DBPedia (0.81%), and SciPhi on TREC (0.78%). This indicates that HIDO can consistently find the order that gives the best performance. (3) **Runner-up on non-optimal datasets**: In those cases that HIDO does not perform the best, the results are still comparable to the best-performing baseline.

## 5.3 Ablation Study

We address **RQ2** by examining four variants of our HIDO: (1) **HIDO-NC**: tests the utility of clustering by randomly assigning samples to clusters. We maintain the same number of clusters and demonstrations per cluster as in the original model. (2) **HIDO-NIntra**: randomly selects demonstration orders within clusters while keeping all other components the same. (3) **HIDO-NInter** randomly selects an inter-cluster order after intra-cluster optimization as the optimal inter-cluster order. (4) **HIDO-ND**: removes the dynamic update scheme for the score function. It outputs the best demonstration order after one iteration. From Fig. 3 (a), we observe that removing each component causes performance degradation. Specifically, we have the following observations: (1) HIDO-NC has the largest difference, indicating that grouping the samples based on distance allows HIDO to find the best order while maintaining efficiency. (2) HIDO-ND has relatively small increase, which implies that HIDO is able to find the best order within a small number of optimization iterations. (3) HIDO-NInter and HIDO-NIntra have similar impacts on the performance. This highlights the significance of our method in finding the best order. See more results in Appendix F.2.

## 5.4 Parameter Sensitivity

Here, we address **RQ3**. Although our model has numerous hyperparameters, we focus our analysis on two we consider most significant: the number of clusters $k$ and the maximum number of optimization iterations $l$. Fig. 3 (b) illustrates our model's performance with varying $k$ and $l$ on the TREC and MPQA datasets using the Sciphi model. We observe that performance improves as $l$ increases, indicating that more iterations of HIDO tend to produce better-performing demonstration orders. Regarding the number of clusters, we find that performance peaks at $k = 2$ for MPQA and $k = 3$ for TREC. This suggests that different datasets require specific cluster numbers for best performance.

## 6 Related Work

**Many-Shot In-Context Learning.** With the expanded context window of developed LLMs, the models can process a larger number of demonstrations within a single prompt, resulting in further research observing the effect of large number of demonstrations (i.e. more than 50) on ICL [1, 12, 15, 2, 25]. [15] develop a long-range language model EVALM that achieves higher accuracy when using many shot ICL; however, the model cannot maintain the same performance consistently, indicating that ICL-DOI still exists. Some emprirical results from [1] provides early evidence for many-shot demonstration order sensitivity by showing how one order that gives the best performance on one subset of a dataset can perform poorly on a different subset of the same original dataset.

**Optimization Techniques for Vast Permutation Spaces.** Finding optimal orderings in large permutation spaces is not unique to ICL. This problem has been studied in various domains. Traditional approaches like simulated annealing [14] and genetic algorithms [35] are applied to similar combinatorial optimization problems. However, these methods often struggle with the scale of permutations encountered in ICL scenarios. Recent work in combinatorial optimization introduces hierarchical and decomposition-based approaches to tackle large-scale permutation problems [10, 22, 29]. For instance, [29] proposes a hierarchical optimization framework for solving large-scale traveling salesman problems, demonstrating the effectiveness of dividing the problem into manageable sub-problems. Enlightened by those ideas, we tackle specific challenges of ICL demonstration ordering.

## 7 Conclusion

We take the initial step on demonstration order instability in many-shot in-context learning. We first propose a score function, ICD-OVI, with solid information theoretical foundation for evaluating demonstration orders. We subsequently design an efficient hierarchical optimization framework `HIDO` that navigates the vast permutation space while maintaining computational feasibility. Extensive experiments verify that `HIDO` achieves significant performance gains across diverse tasks. Our information-theoretic approach provides both theoretical guarantees and practical benefits, opening new avenues for unleashing ICL performance.

## 8 Acknowledgments

This work is supported in part by the National Science Foundation (NSF) under grants IIS-2006844, IIS-2144209, IIS-2223769, IIS-2331315, CNS-2154962, BCS-2228534, and CMMI-2411248, the Office of Naval Research (ONR) under grant N000142412636, and the Commonwealth Cyber Initiative (CCI) under grant VV-1Q25-004.

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

# Appendix

## A Limitations

In this work, several limitations exist that should be acknowledged for a balanced understanding of the results and methodology. First, while the Hierarchical Demonstration Order Optimization (HIDO) framework effectively reduces the search space for many-shot in-context learning (ICL), its reliance on clustering introduces an additional layer of complexity that may not always generalize well to all datasets or language models. The clustering process itself, especially with a limited number of clusters, may not capture intricate interdependencies between demonstrations. Furthermore, although the dynamic update mechanism improves the accuracy of the score function, it also increases the overall computational cost, particularly when applied to very large datasets or when running a high number of optimization iterations.

Additionally, the current framework assumes that performance improvements arise primarily from the optimized demonstration order, but factors such as the inherent instability of large language models (LLMs) across varying contexts might also contribute to observed fluctuations. Finally, the probing set generation step introduces potential noise, and while the system attempts to mitigate this through iterative updates, inaccuracies in probing may still affect the final demonstration order selection.

## B Broader Impact

This work on Hierarchical Demonstration Order Optimization (HIDO) for many-shot in-context learning has significant potential societal implications. Positively, by improving the performance and reliability of large language models across diverse domains, HIDO could enhance AI applications in education, healthcare, and scientific research, making these systems more accessible and effective for users without extensive prompt engineering expertise. However, potential negative impacts include reinforcing existing biases in training data through optimized demonstration orders, increasing compute requirements for determining optimal orderings (raising environmental and resource accessibility concerns), and potentially widening capability gaps between organizations with resources to implement such optimization techniques and those without. As many-shot in-context learning becomes more widely deployed, careful consideration should be given to monitoring how optimization techniques like HIDO affect fairness, bias, and resource distribution in AI systems.

## C Licenses for existing assets

**Datasets:**

- AGNews [46]: publicly available for research use.
- CB [7]: released under MIT License.
- CR [11]: publicly available for research use.
- DBPedia [46]: available under Creative Commons Attribution-ShareAlike License.
- MPQA [39]: available under GNU General Public License.
- MR [30]: publicly available for research use.
- RTE [6]: publicly available for research use.
- SST-5 [34]: released under Stanford CoreNLP License.
- TREC [37]: released under Creative Commons Attribution 4.0 license.

**Models:**

- GPT-3.5-Turbo and GPT-4o-Mini [26, 27]: used under the OpenAI API Terms of Use.
- SciPhi-Mistral-7B-32k [33]: released under Apache 2.0 License.
- Zephyr-7b-beta [36]: Cite Huggingface [14], released under MIT License.
- LlaMa-3-8B-Instuct-Gradient-1048k [32]: released under Llama 3 Community License.

We have added this license information in Section 5.1 and ensured proper attribution throughout the paper. All assets are used in accordance with their respective licenses and terms of use.

# D   Complete Development of ICD-OVI Metric

Enlightened by $\mathcal{V}$-usable information, our ICD-OVI, measures the usable information that an LLM can capture from ordered demonstrations $\Pi(\mathcal{D})$. First, we define the predictive family corresponding to the ordered demonstrations $\Pi(\mathcal{D})$ as

$$\mathcal{V}_\Pi := \{P_{\text{LLM}}(\cdot|\Pi(\mathcal{D}) \oplus q)|q \in \mathcal{Q}_P\} \cup \{P_{\text{LLM}}(\cdot|q)|q \in \mathcal{Q}_P\}, \tag{6}$$

where $\mathcal{Q}_P$ represents the set of all possible queries in the sample space of input demonstrations' data distribution $P$, and $\{P_{\text{LLM}}(\cdot|q)|q \in \mathcal{Q}_P\}$ is added to satisfy the optimal ignorance requirement for a predictive family [42]. Then, ICD-OVI, the information that the model can capture from $\Pi(\mathcal{D})$, can be defined as the expected information the model with predictive family $\mathcal{V}_\Pi$ can capture from query random variable (r.v.) $Q$ for predicting label r.v. $A$, i.e.,

$$\begin{aligned} \text{ICD-OVI} &= H_{\mathcal{V}_\Pi}(A) - H_{\mathcal{V}_\Pi}(A|Q), \\ &= \inf_{f \in \mathcal{V}_\Pi} \mathbb{E}_{q,a \sim \mathcal{D}}[-\log f[\emptyset](a)] - \inf_{f \in \mathcal{V}_\Pi} \mathbb{E}_{q,a \sim \mathcal{D}}[-\log f[q](a)], \\ &= \mathbb{E}_{(q,a) \sim P}[\log_2 P_{\text{LLM}}(a|\Pi(\mathcal{A}) \oplus \emptyset) - \log_2 P_{\text{LLM}}(a|\Pi(\mathcal{D}) \oplus q)], \end{aligned} \tag{7}$$

where $\Pi(\mathcal{A}) := \bigoplus_{i=1}^n \mathcal{T}(\emptyset, a_{\pi(i)})$. The third equation follows the definition of in-context $\mathcal{V}$-information from Eq. 1 of [20]. Practically, denoting $P_{\text{LLM}}^i(\hat{a}) := P_{\text{LLM}}(\hat{a}|\hat{q}_i)$, we may approximate the Eq. 7 with the probing samples $\hat{D}$ generated by LLM with

$$\frac{1}{|\hat{D}|} \Sigma_i (-\log_2 P_{\text{LLM}}^{\Pi,i}(\hat{a}) + \log_2 P_{\text{LLM}}^i(\hat{a})). \tag{8}$$

However, Eq. 8 involves the LLM-generated labels $\hat{a}$s for the probing samples, which can be factually incorrect. Utilizing those incorrect labels may lead to bias in the computation of ICD-OVI. Fortunately, the theory of $V$-usable information [9, 20] provide a effective tool called point-wise $\mathcal{V}$-informationn threshold (*PVI threshold*) which assists deciding if one generated sample label is reliable. Here, PVI is defined as

$$\text{PVI}_{(\hat{q},\hat{a})}^{\Pi(\mathcal{D})} = -\log_2 P_{\text{LLM}}(\hat{a}|\Pi(\mathcal{D}) \oplus \hat{q}) + \log_2 P_{\text{LLM}}(\hat{a}|\Pi(\mathcal{A}) \oplus \hat{q}). \tag{9}$$

By Eq. 9, the ICD-OVI is the mean of PVIs for all probing samples $\hat{\mathcal{D}}$. Built upon PVI, the PVI threshold is a scalar characterizing the likelihood of the correctness of the sample label. Specifically, when the PVI of a probing sample $(\hat{q}, \hat{a})$ is smaller than a constant $\tau$, the label $\hat{a}$ is possibly incorrect; otherwise, the label $\hat{a}$ is highly likely to be correct for query $\hat{q}$. Actually, the fact of the existence of a PVI threshold is extensively validated by [9] and [20] in multiple LLMs and datasets of various semantic scenarios.

With the aid of the PVI threshold, we can address the potential bias caused by incorrect LLM-generated labels. Specifically, for a probing sample $(\hat{q}, \hat{a})$, we first calculate its PVI; if it is higher than a predefined $\mathcal{V}$-information threshold $\tau$, then we adopt the PVI of the sample $(\hat{q}, \hat{a})$ into the ICD-OVI calculation of ordered demonstrations $\Pi(\mathcal{D})$. Otherwise, we relax the PVI to its expectation for labels set $\{a|a \in \mathcal{A}\}$, i.e.,

$$\text{EPVI}_{(\hat{q},\hat{a})}^{\Pi(\mathcal{D})} = \Sigma_{a \in \mathcal{A}}[-P_{\text{LLM}}^{\Pi,\hat{q}}(a)\log_2 P_{\text{LLM}}^{\Pi,\hat{q}}(a) + P_{\text{LLM}}^{\hat{q}}(a)\log_2 P_{\text{LLM}}^{\hat{q}}(a)]. \tag{10}$$

Conclusively, by denoting point-wise ICD-OVI (PICD-OVI) as

$$\text{PICD-OVI}_{(\hat{q},\hat{a})}^{\Pi(\mathcal{D})} = \mathbb{I}(\text{PVI}_{(\hat{q},\hat{a})} \geq \tau)\text{PVI}_{(\hat{q},\hat{a})} + \mathbb{I}(\text{PVI}_{(\hat{q},\hat{a})} < \tau)\text{EPVI}_{(\hat{q},\hat{a})}, \tag{11}$$

our ICD-OVI can be approximated as

$$\text{ICD-OVI}(\Pi(\mathcal{D})) \approx \frac{1}{|\hat{D}|} \Sigma_{(\hat{q},\hat{a})} \text{PICD-OVI}_{(\hat{q},\hat{a})}. \tag{12}$$

Thus, our proposed ICD-OVI can effectively estimate the $V$-usable information despite noisy labels.

# E   Theorems and Proofs

**Lemma 1.** *Let $f(x_1, \ldots, x_n) = \sum_{i=1}^{n} x_i \log x_i$ be defined for $x_i > 0$, with the constraint $\sum_{i=1}^{n} x_i = c$, where $0 < c < \frac{1}{e}$. Then:*

1. *$f$ reaches its minimum when all $x_i$ are equal, i.e., $x_i = \frac{c}{n}$ for all $i$.*

2. *$f$ reaches its maximum when one $x_i$ equals $c$ and the rest are zero.*

*Proof.* We will use the method of Lagrange multipliers.

Let $g(x_1, \ldots, x_n) = \sum_{i=1}^{n} x_i - c = 0$ be our constraint. The Lagrangian is:

$$L(x_1, \ldots, x_n, \lambda) = \sum_{i=1}^{n} x_i \log x_i - \lambda \left( \sum_{i=1}^{n} x_i - c \right)$$

We set the partial derivatives to zero:

$$\frac{\partial L}{\partial x_i} = \log x_i + 1 - \lambda = 0 \quad \text{for } i = 1, \ldots, n$$

$$\frac{\partial L}{\partial \lambda} = \sum_{i=1}^{n} x_i - c = 0$$

From $\frac{\partial L}{\partial x_i} = 0$, we get:

$$x_i = e^{\lambda - 1}$$

This shows that all $x_i$ are equal at the critical points.

**Minimum Point:** When all $x_i$ are equal, let $x_i = \frac{c}{n}$ for all $i$. The function value is:

$$f\left( \frac{c}{n}, \ldots, \frac{c}{n} \right) = c \log \frac{c}{n}$$

**Maximum Point:** Consider $x_1 = c$ and $x_i = 0$ for $i > 1$. The function value is:

$$f(c, 0, \ldots, 0) = c \log c$$

To show that $f\left( \frac{c}{n}, \ldots, \frac{c}{n} \right) < f(c, 0, \ldots, 0)$, we need to prove:

$$c \log \frac{c}{n} < c \log c$$

This is equivalent to $frac{c}{n} < c$, which is true for $n > 1$ and $c > 0$. Therefore, we have shown that the minimum occurs when all $x_i = \frac{c}{n}$, and the maximum occurs when one $x_i = c$ and the rest are zero. $\square$

**Theorem 1** *We assume that given a LLM, a probing sample $(\hat{q}, \hat{a})$ and an ordered demonstration text $\Pi(\mathcal{D})$,*

- *When $PVI_{(\hat{q}, \hat{a})}^{\Pi(\mathcal{D})} \geq \tau$, then $\hat{a} = a^*$, where the $a^*$ is the ground-truth label corresponding to the generated query $\hat{q}$.*

- *The LLM predict the label $\hat{a}$ with the highest probability when query by $\hat{q}$ with $\Pi(\mathcal{D})$ as its context, i.e., $P(\hat{a}|\Pi(\mathcal{D}) \oplus \hat{q}) = \arg\max_{a \in \mathcal{A}} P(a|\Pi(\mathcal{D}) \oplus \hat{q})$.*

- *Assume that for any two ordered demonstration texts $\Pi_1(\mathcal{D})$ and $\Pi_2(\mathcal{D})$, the $P_{LLM}(a|\Pi_1(\mathcal{A}) \oplus \emptyset) = P_{LLM}(a|\Pi_2(\mathcal{A}) \oplus \emptyset)$ for all $a \in \mathcal{A}$.*

*Without loss of generalizability, for any two ordered demonstrations $\Pi_1(\mathcal{D})$ and $\Pi_1(\mathcal{D})$, there is a $\epsilon(\frac{1}{e} \leq \epsilon \leq 1)$ such that $P(\hat{a}|\Pi_i(\mathcal{D}) \oplus \hat{q}) > 1 - \epsilon$. We additionally assume that when $PVI_{(\hat{q}, \hat{a})}^{\Pi(\mathcal{D})} < \tau$:*

- *The $a^*$ is the second most probable label given by the LLM when prompted by query $\hat{q}$ with any ordered demonstration context $\Pi(\mathcal{D})$, i.e., $P(a^*|\Pi(\mathcal{D}) \oplus \hat{q}) = \arg\max_{a \in \mathcal{A} \setminus \{\hat{a}\}} P(a|\Pi(\mathcal{D}) \oplus \hat{q})$; we write $P(a^*|\Pi_i(\mathcal{D}) \oplus \hat{q}) = \lambda_i \epsilon$, where $0 \le \lambda_i \le 1$, $i \in \{1, 2\}$.*

- *By symmetry, we only consider the case $\lambda_1 < \lambda_2$. In this case, we assume that $\frac{1}{2} - \delta < \lambda_1 < \frac{1}{2} + \delta$ ($\delta$ is a constant) such that*

$$(\lambda_1 \epsilon) \log \lambda_1 \epsilon + (1 - \lambda_1 \epsilon) \log (1 - \lambda_1 \epsilon) < \epsilon \log \epsilon - (2 - \lambda_1) \epsilon. \tag{13}$$

  *Meanwhile, we require $\lambda_2 - \lambda_1 > (1 - \frac{1}{\log(n-2)})(1 - \lambda_1)$.*

*With the assumptions above, if*

$$PICD\text{-}OVI^{\Pi_1(\mathcal{D})}_{(\hat{q},\hat{a})} > PICD\text{-}OVI^{\Pi_2(\mathcal{D})}_{(\hat{q},\hat{a})}, \tag{14}$$

*then we have*

$$PVI^{\Pi_1(\mathcal{D})}_{(\hat{q},a^*)} > PVI^{\Pi_2(\mathcal{D})}_{(\hat{q},a^*)}. \tag{15}$$

*Therefore, if $\Pi_1(\mathcal{D})$ is more performant demonstration order than $\Pi_2(\mathcal{D})$, i.e., Eq. 15 establish for any probing sample $(\hat{q}, \hat{a})$, then*

$$ICD\text{-}OVI(\Pi_1(\mathcal{D})) > ICD\text{-}OVI(\Pi_2(\mathcal{D})). \tag{16}$$

*Proof.* First, in the case that $PVI^{\Pi(\mathcal{D})}_{(\hat{q},\hat{a})} \ge \tau$, by Assumption 1, we have $\hat{a} = a^*$. Therefore, we have

$$\text{PICD-OVI}^{\Pi(\mathcal{D})}_{\hat{q},\hat{a}} = P(\hat{a}|\Pi(\mathcal{D}) \oplus \hat{q}) - P(\hat{a}|\Pi(\mathcal{A}) \oplus \emptyset) = \text{PVI}^{\Pi(\mathcal{D})}_{\hat{q},\hat{a}} = \text{PVI}^{\Pi(\mathcal{D})}_{\hat{q},a^*}. \tag{17}$$

Eq. 17 enforces the establishment of Eq. 15.

Next, in the case where $\text{PVI}^{\Pi(\mathcal{D})}_{(\hat{q},\hat{a})} < \tau$, with Assumption 3, it suffices to prove that $|\lambda_1 \epsilon \log \lambda_1 \epsilon| \ge |\lambda_2 \epsilon \log \lambda_2 \epsilon|$ gives rise to

$$|\lambda_1 \epsilon \log \lambda_1 \epsilon + \Sigma_{\Sigma_i x_i = (1-\lambda_1)\epsilon} x_i \log x_i + x_{\hat{a},1}| \ge |\lambda_2 \epsilon \log \lambda_2 \epsilon + \Sigma_{\Sigma_i x_i = (1-\lambda_1)\epsilon} x_i \log x_i + x_{\hat{a},2}|. \tag{18}$$

Now, by utilizing the Assumption 5, we claim that Eq. 18 establish, thus the theorem is proved.

To prove Eq. 18, we start from the known inequivality

$$\lambda_2 - \lambda_1 > (1 - \frac{1}{\log(n-2)})(1 - \lambda_1). \tag{19}$$

For simplicity, we represent $\lambda_2 - \lambda_1$ as $\Delta$ in the following texts. We rewrite the Eq. 19 as

$$\begin{aligned}
\Delta &> \frac{1 + 1/\epsilon \log e^{-\epsilon(1-\lambda_1)-\epsilon+\log 2/2}}{\log(n-2)} + (1 - \lambda_1), \\
&= \frac{1}{\epsilon} \Big[ \frac{\epsilon(\log \epsilon - \log 2) + (\epsilon \log 2 - \epsilon(2 - \lambda_1))}{\log(n-2)} \Big] - \frac{\log \epsilon}{\log(n-2)} + (1 - \lambda_1) + \frac{1}{\log(n-2)}.
\end{aligned} \tag{20}$$

By Assumption 5, we substitute terms appears in Eq. 20 with left hand side (LHS) of Eq. 13 and $\log[1 - \lambda_1 \epsilon] > \log[(1 - \lambda_1)\epsilon]$, further relax the bound as

$$\begin{aligned}
\Delta &> \lambda_1 \frac{\log \lambda_1 \epsilon}{\log(n-2)} - \frac{\log \epsilon}{\log(n-2)} + (1 - \lambda_1) + \frac{1 - \lambda_1}{\log(n-2)} \log[(1 - \lambda_1)\epsilon] + \frac{1}{\log(n-2)} \\
&= -\frac{1}{\epsilon \log(n-2)} \{ -\lambda_1 \epsilon \log \lambda_1 \epsilon + \epsilon \log \epsilon - [(1 - \lambda_1)\epsilon] \log(n-2) - (1 - \lambda_1)\epsilon \log[(1 - \lambda_1)\epsilon] - \epsilon \}.
\end{aligned} \tag{21}$$

By multipling $\epsilon \log(n-2)$ to both sides of the inequivality, we have

$$-\lambda_1 \epsilon \log(\lambda_1 \epsilon) + \epsilon \log \epsilon - [\log(n-2)](1 - \lambda_1 - \Delta)\epsilon - (1 - \lambda_1)\epsilon \log(1 - \lambda_1)\epsilon - \epsilon > 0. \tag{22}$$

Eq. 22 is equivalent to

$$-\lambda_1 \epsilon \log \lambda_1 \epsilon + \log \epsilon (\lambda_1 + \Delta)\epsilon + (n-2)\frac{(1 - \lambda_1 - \Delta)\epsilon}{n-2} \log \frac{\epsilon}{n-2} - (1 - \lambda_1)\epsilon \log(1 - \lambda_1)\epsilon - \epsilon > 0. \tag{23}$$

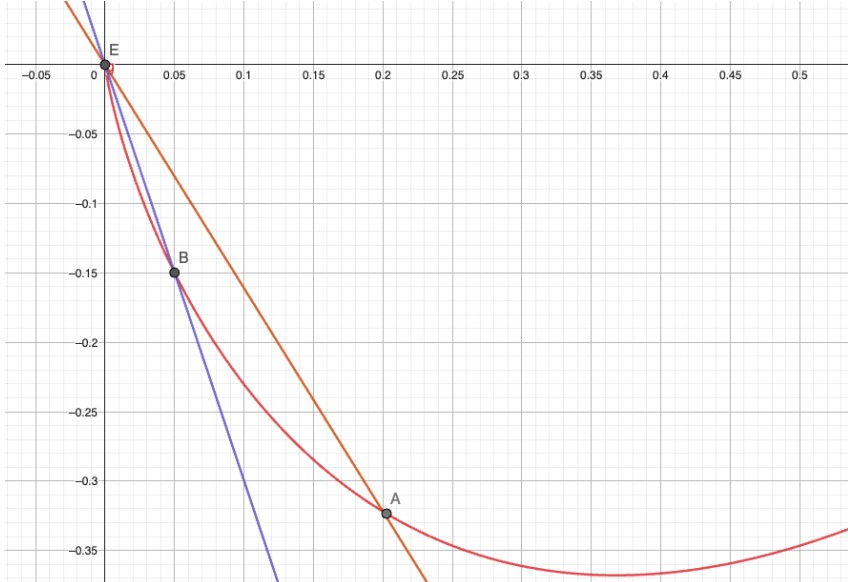

Figure 4: Illustration of the observation of Eq. 24 and Equ 25. The red, orange, and blue curves are $x \log x$, $\log \epsilon x$ and $\log \frac{\epsilon}{n-2} x$ (where $n = 6$ and $\epsilon = 0.2$), respectively. It is clear that $x \log x \leq \log \epsilon x$ between point $E$ and $A$; $x \log x \leq \log \frac{1}{n-2} \epsilon x$ between point $E$ and $B$.

Now, we observe that since $\lambda_1 + \Delta = \lambda_2 < 1$, thus $(\lambda_1 + \Delta)\epsilon < \epsilon$. Therefore

$$(\lambda_1 + \Delta)\epsilon \log (\lambda_1 + \Delta)\epsilon \leq -\log \epsilon (\lambda_1 + \Delta)\epsilon. \tag{24}$$

Here, the $\log \epsilon$ is the slope of the linear function composed by $(0,0)$ and $(\epsilon, \epsilon \log \epsilon)$. Analogously, we have

$$\frac{1 - \lambda_1 - \Delta}{\epsilon} \log \frac{1 - \lambda_1 - \Delta}{n-2} \epsilon \leq \log \frac{\epsilon}{n-2} \frac{(1 - \lambda_1 - \Delta)\epsilon}{n-2}. \tag{25}$$

By substituting the terms of RHS of Equ 24 and Equ 25 appeared in Equ 23 with the LHS of Equ 24 and Equ 25, we further relax our inequivality as

$$-\lambda_1 \epsilon \log \lambda_1 \epsilon + (\lambda_1 + \Delta)\epsilon \log (\lambda_1 + \Delta)\epsilon + (1 - \lambda_1 - \Delta)\epsilon \log \left(\frac{1 - \lambda_1 - \Delta}{n-2} \epsilon\right) -$$
$$(1 - \lambda_1)\epsilon \log (1 - \lambda_1)\epsilon + (1 - \epsilon) \log 1 - \epsilon > 0. \tag{26}$$

We now rearrange the Eq. 26 and substitute $\lambda_1 + \Delta$ with $\lambda_2$, we have

$$-\lambda_1 \epsilon \log \lambda_1 \epsilon - (1 - \lambda_1)\epsilon \log (1 - \lambda_1)\epsilon >$$
$$-(\lambda_2 \epsilon) \log (\lambda_2 \epsilon) - (1 - \lambda_2)\epsilon \log \left(\frac{1 - \lambda_1 - \Delta}{n-2} \epsilon\right) - (1 - \epsilon) \log (1 - \epsilon). \tag{27}$$

We observe that, by Lemma 1, we have that

$$\min_{(x_1, \ldots, x_{n-2})} \Sigma_{\Sigma x_i = (1 - \lambda_2)\epsilon} x_i \log x_i = (1 - \lambda_2)\epsilon \log \left(\frac{1 - \lambda_2}{n-2} \epsilon\right),$$
$$\max_{(x_1, \ldots, x_{n-2})} \Sigma_{\Sigma x_i = (1 - \lambda_1)\epsilon} x_i \log x_i = (1 - \lambda_1)\epsilon \log (1 - \lambda_1 \epsilon). \tag{28}$$

In other words,

$$\max_{(x_1, \ldots, x_{n-2})} |\Sigma_{\Sigma x_i = (1 - \lambda_2)\epsilon} x_i \log x_i| = -(1 - \lambda_2)\epsilon \log \left(\frac{1 - \lambda_2}{n-2} \epsilon\right),$$
$$\min_{(x_1, \ldots, x_{n-2})} |\Sigma_{\Sigma x_i = (1 - \lambda_1)\epsilon} x_i \log x_i| = -(1 - \lambda_1)\epsilon \log (1 - \lambda_1 \epsilon). \tag{29}$$

Besides, it is direct to show that

$$(1 - \epsilon) \log (1 - \epsilon) \leq x_{\hat{a},i} \log x_{\hat{a},i} \leq 0, \tag{30}$$

i.e.,
$$-(1 - \epsilon) \log (1 - \epsilon) \geq |x_{\hat{a},i} \log x_{\hat{a},i}| \geq 0, \tag{31}$$
Hence, we rewrite the Eq. 27 to

$$|\lambda_1 \epsilon \log \lambda_1 \epsilon| + \min_{(x_1,\ldots,x_{n-2})} |\Sigma_{\Sigma x_i = (1-\lambda_1)\epsilon} x_i \log x_i| + \min |x_{\hat{a},1} \log x_{\hat{a},1}| >$$
$$|(\lambda_2 \epsilon) \log (\lambda_2 \epsilon)| + \max_{(x_1,\ldots,x_{n-2})} |\Sigma_{\Sigma x_i = (1-\lambda_2)\epsilon} x_i \log x_i| + \max |x_{\hat{a},2} \log x_{\hat{a},2}|. \tag{32}$$

Therefore, we are able to write that

$$|\lambda_1 \epsilon \log \lambda_1 \epsilon + \Sigma_{\Sigma_i x_i = (1-\lambda_1)\epsilon} x_i \log x_i + x_{\hat{a},1}| \geq |\lambda_2 \epsilon \log \lambda_2 \epsilon + \Sigma_{\Sigma_i x_i = (1-\lambda_1)\epsilon} x_i \log x_i + x_{\hat{a},2}|, \tag{33}$$

which is exactly Eq. 18. $\qquad\square$

**Theorem 2.** Randomly flipping $K$ entries from a sequence of length $N$ will always keep the rank correlation within a range characterized by the lower bound $1 - 6 \sum_{i=1}^{K} (a_i - a_{K+1-i})^2 / N(N^2 - 1)$ and upper bound 1. Here $a_i$ is the original position index of the $i$-th perturbed element. The lower bound is achieved with a probability of $1/K!$ when the perturbed sequence is the reverse of the original sequence. The upper bound is achieved with a probability of $1/K!$ when the perturbed sequence is identical to the original sequence.

To prove the above theorem, we first present the lemma:

**Lemma 2.** *Given a list of $N$ integers $\{a_1, a_2, \ldots, a_N\}$ with $a_i < a_{i+1}, i = 1, 2, \ldots, N - 1$ and its random perturbation $\{a_1^*, a_2^*, \ldots, a_N^*\}$, the maximum value of $\sum_{i=1}^{N} (a_i - a_i^*)^2$ is achieved by reversing the list, i.e., $a_i^* = a_{N+1-i}$.*

*Proof.* To prove that the maximum value of the sum:

$$S = \sum_{i=1}^{N} (a_i - a_i^*)^2$$

is achieved by reversing the list $\{a_i^*\}_{i=1}^N$, we need to show that this arrangement maximizes the squared differences between the original list $\{a_i\}_{i=1}^N$ and the perturbed list $\{a_i^*\}_{i=1}^N$, where $a_i^*$ is the perturbed element in the $i$-th position.

We know that
$$a_1 < a_2 < \cdots < a_N.$$
Considering the sum $S = \sum_{i=1}^{N} (a_i - a_i^*)^2$, each term in this sum is of the form $(a_i - a_i^*)^2$, which measures how far apart $a_i$ and $a_i^*$ are. Thus, to maximize the sum, we need to maximize each individual squared difference $(a_i - a_i^*)^2$.

The largest possible difference between any two elements of the list $\{a_i\}_{i=1}^N$ occurs when the largest element $a_N$ is paired with the smallest element $a_1$, the second largest element $a_{N-1}$ is paired with the second smallest element $a_2$, and so on. In other words, the maximum possible difference occurs when $a_i^* = a_{N+1-i}$ for all $i$. This arrangement is precisely the reverse of the original list.

To prove that reversing the list maximizes the sum, we propose to prove that when swapping any two elements in the perturbed list, the sum will always decrease. Suppose we swap two elements $a_p^*$ and $a_q^*$ (with $p < q$, without loss of generality) in the reversed list. Before the swap, the contributions to the sum from the two positions are:

$$(a_p - a_p^*)^2 + (a_q - a_q^*)^2.$$

After swapping $a_p^*$ and $a_q^*$, the new contributions become:

$$(a_p - a_q^*)^2 + (a_q - a_p^*)^2.$$

The change in the sum, $\Delta S$, is the difference between these two expressions:

$$\Delta S = \left((a_p - a_q^*)^2 + (a_q - a_p^*)^2\right) - \left((a_p - a_p^*)^2 + (a_q - a_q^*)^2\right).$$

We expand these terms as follows:

- Before the swap:

$$(a_p - a_p^*)^2 + (a_q - a_q^*)^2 = (a_p - a_{N+1-p})^2 + (a_q - a_{N+1-q})^2$$

- After the swap:

$$(a_p - a_q^*)^2 + (a_q - a_p^*)^2 = (a_p - a_{N+1-q})^2 + (a_q - a_{N+1-p})^2$$

Because $a_p < a_q$ and the list is ordered, swapping two elements in the reversed list *decreases* the squared differences, leading to a decrease in the sum $S$. Thus, reversing the list maximizes the absolute differences $|a_i - a_i^*|$ for all $i$, and any deviation from the reversed order will result in a smaller sum. $\square$

With this lemma, now we prove Theorem 2.

*Proof.* Given two ranking sequences $\{s_i\}_{i=1}^N$ and $\{s_i^*\}_{i=1}^N$, the Spearman's rank correlation coefficient is represented as follows:

$$\rho = 1 - \frac{6 \sum_{i=1}^N (s_i - s_i^*)^2}{N(N^2 - 1)}. \tag{34}$$

In our case, one ranking sequence is obtained by perturbing $K$ elements in another ranking sequence. Denote the selected elements as $\{a_i\}_{i=1}^K$, and the elements after perturbation as $\{a_i^*\}_{i=1}^K$

according to Lemma 2, we know the maximum value of $\sum_{i=1}^K (a_i - a_i^*)^2$ is achieved when $a_i^* = a_{K+1-i}$.

For other elements that are not perturbed satisfy that their $d_i$ equals 0. Therefore, the Spearman's rank correlation coefficient reaches the minimum value:

$$\rho_{\min} = 1 - \frac{6 \sum_{i=1}^K (a_i - a_{K+1-i})^2}{N(N^2 - 1)}. \tag{35}$$

Similarly, the maximum value is $\rho_{\max} = 1$ when the perturbed sequence is exactly the same as the original sequence. Since each perturbation has an equal probability, and there are $K!$ different perturbations, we know the probabilities are both $1/K!$. $\square$

## F   Supplementary Experiments

### F.1   Implementation Details

Our conduct experiments using a system equipped with four NVIDIA A100 80GB PCIe GPUs. The system ran NVIDIA driver version 550.54.14 and CUDA 12.4. We implement the project with Python, mainly relying on the PyTorch [31] and Transformers [40] packages for the implementation.

### F.2   Ablation Experiment Results

Please see the supplementary ablation experiment results in Fig. 5. We have consistent observations from those results with the main paper.

### F.3   Accuracy difference between few-shot (10 shots) and many-shot (150) ICL

We want to confirm that ICL-DOI still exists in many shot ICL. Thus, we randomly select orders with 10 or 150 demonstrations and measure the model accuracy. The following figures present the distribution of model performance under few-shot and many-shot settings on various datasets.

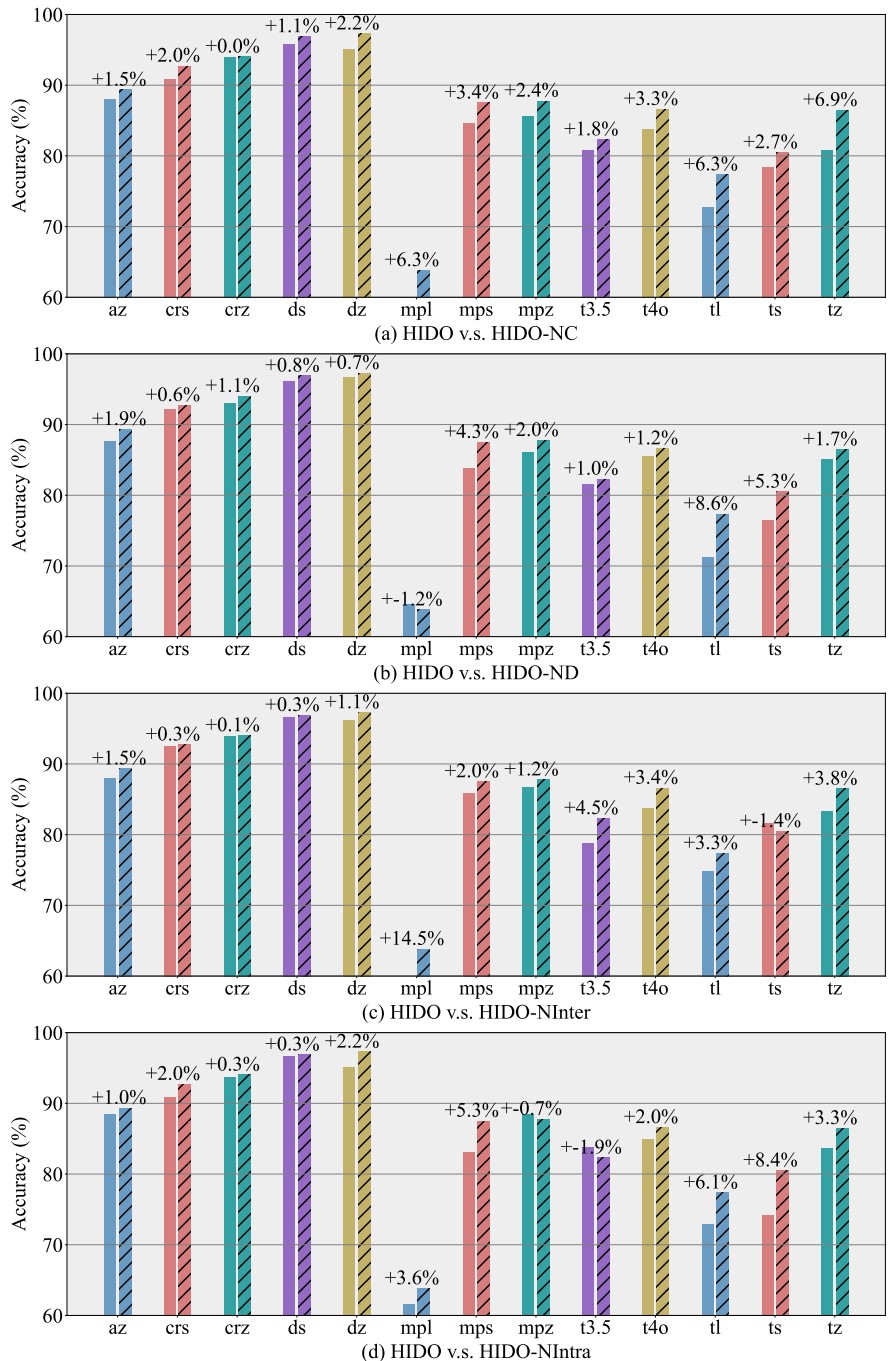

Figure 5: The performance of our proposed HIDO and its variants tested with different LLMs on various datasets. The first one or two characters indicate the dataset (i.e. 't' represents TREC and 'mp' represents 'MPQA'). The remaining characters represent the model (i.e. 'z' represents Zephyr and '3.5' represents GPT-3.5T).

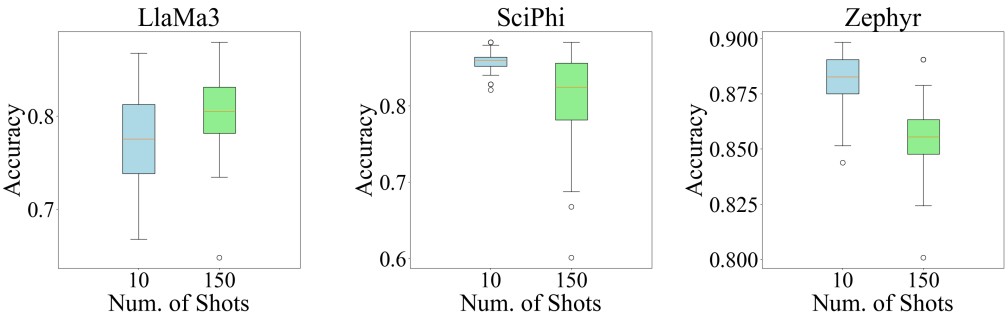

Figure 6: AGNews. Many shot ICL generally improves the best model accuracy (i.e. increases maximum accuracy), which causes the range to be larger.

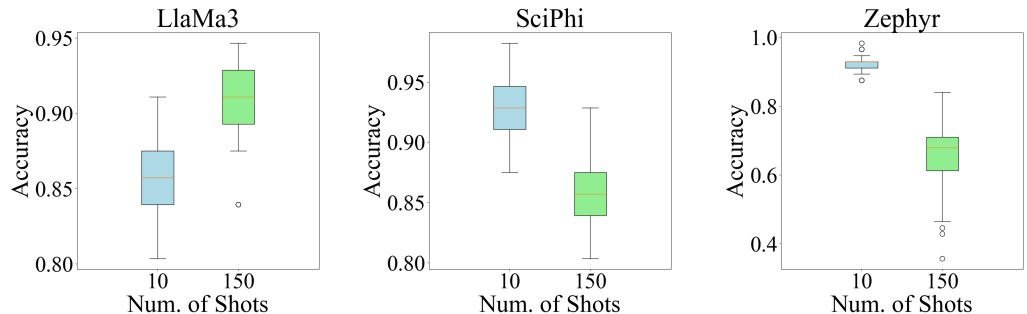

Figure 7: CB. Here, the figure shows that many shot learning causes model performance to degrade. This could be a result of CB having less test samples (56 samples compared to 256 samples for other datasets). Regardless, there is large variance in the results, indicating demonstration order instability.

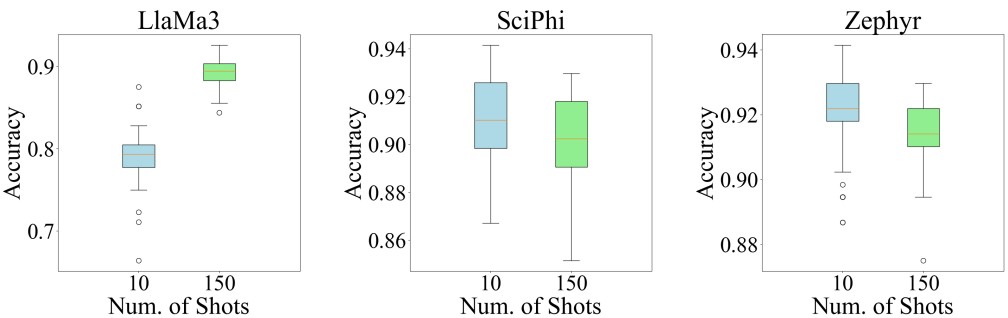

Figure 8: CR. SciPhi and Zephyr exhibit a wider variance in accuracy. In Zephyr, there is an extremely low outlier, emphasizing the importance of order on model performance.

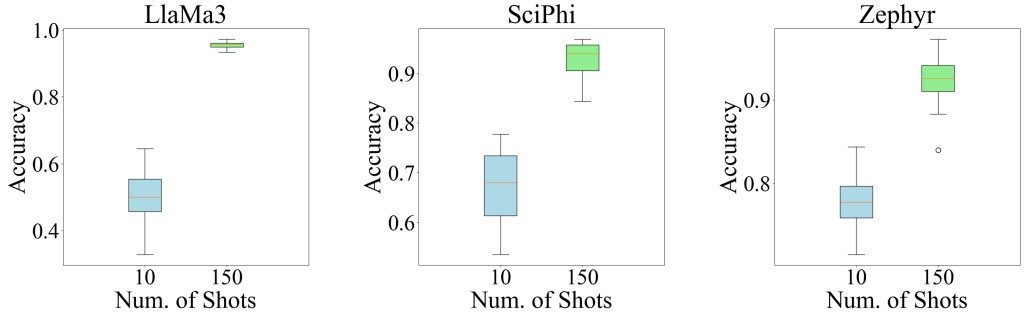

Figure 9: DBPedia. Many shot learning improved model performance for all models; however, for LlaMa3, the variance becomes smaller but stays the same or increases for the other models. Taking a look at DBPedia, the samples in general give more context in comparison to the others, which suggests that LlaMa3 is better at retaining and exploiting the information given from the demonstrations when completing the task of interest.

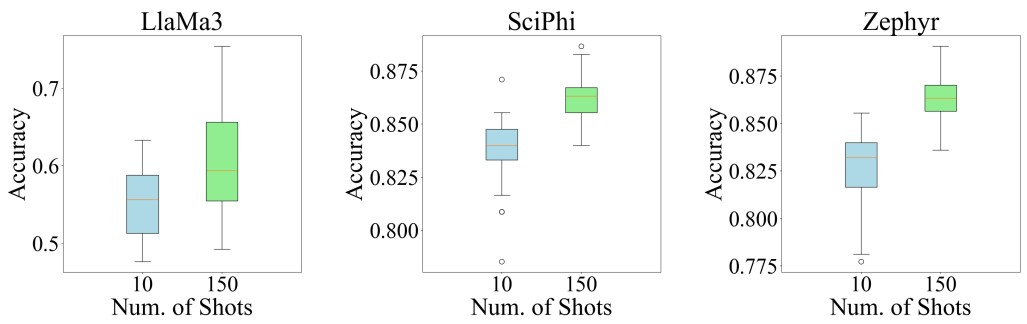

Figure 10: MPQA. Again, many shot ICL improved model accuracy, but also caused the variance to increase in general. LlaMa3 especially exhibits the problem of ICL-DOI with over 25% difference between the best and worst accuracy.

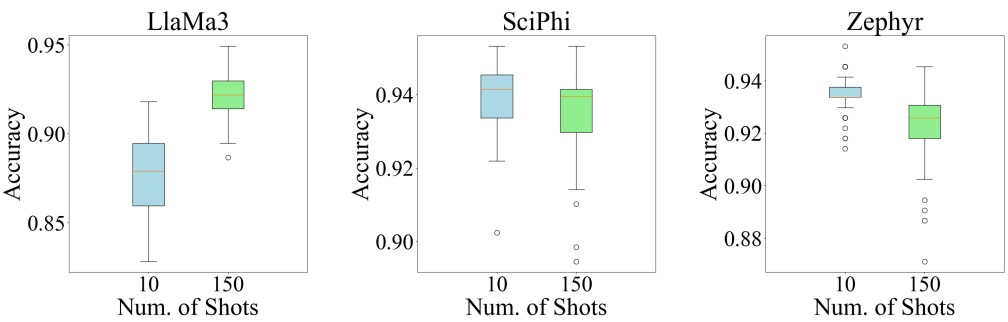

Figure 11: MR. Model performance only improved for LlaMa3, but the other two models illustrate a wider variance. For SciPhi and Zephyr, the model performance under the few-shot and many shot settings is comparable, but in many-shot, the worst accurcy is much lower than that of few-shot performance.

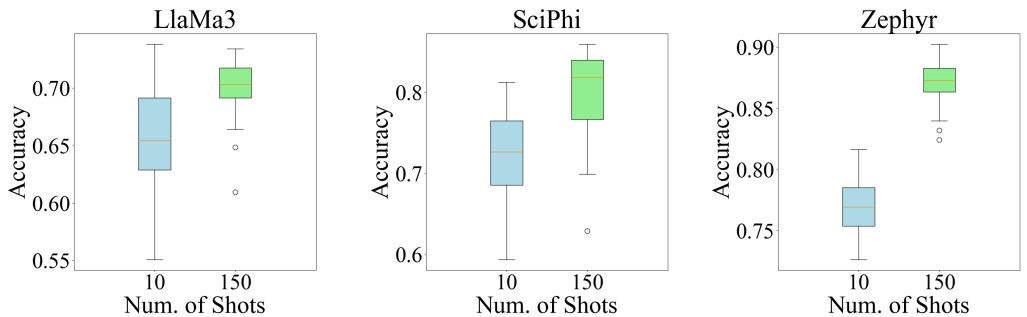

Figure 12: TREC. Increasing the number of demonstrations increased average accuracy for all models, and the variance did not improve much, other than for LlaMa3. LlaMa3 has 8 billion paramters, compared to only 7 billion for the other two models, which means that it has more capability to learn and retain information. This can potentially be the reason for its superior performance against the other two LLMs.

## F.4 Quantitative Analysis of Generated Probing Sets

In the method development, we assume that the demonstration order optimized for answer prediction can also be used for sample generation. Since each additional iteration of HIDO optimizes the order such that it can achieve a higher accuracy, the probing set from the inter-cluster optimization round is generated from the current optimized order. Thus, we can compare the probing set to the original demonstrations, which should be of high quality. Ideally, as the number of iterations increases (i.e. the order becomes more optimized), the distance between the two should decrease (i.e. the quality of the probing set increases). The following figures measure the average $L_2$ norm between the demonstration embeddings and the probing set embeddings generated by various LLMs on different datasets. In general, the experiments support the assumption, presenting a negative trend between iterations and distance.

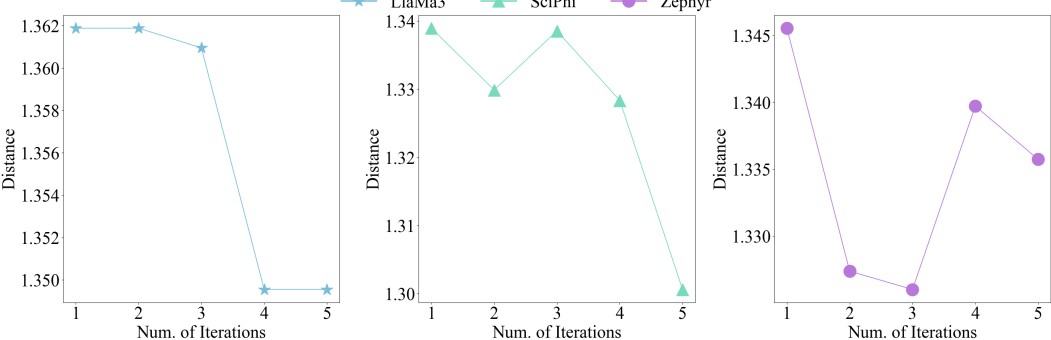

Figure 13: AGNews. Embedding distance for both LlaMa3 and Zephyr consistently decrease as the number of iterations increase; however, Zephyr reaches its optimal at three iterations, and additional iterations will cause the resulting order to deviate, as indicated by the spike at the fourth iteration. SciPhi has a peak at three iterations but decreases after that point.

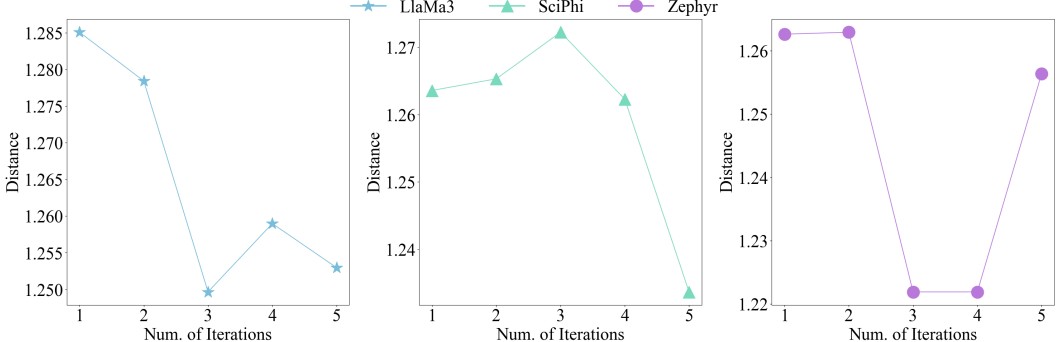

Figure 14: CR. Similar to the previous figure, the probing sets generated by LlaMa3 and Zephyr consistently drop, and SciPhi displays a peak and then a major drop in embedding distance. The figures suggest that after some iterations (i.e. as the order becomes more optimal), the LLM can generate samples close to the original text.

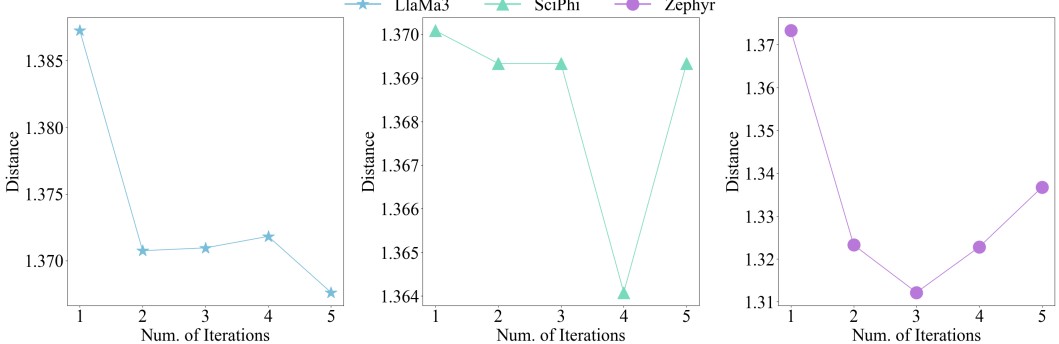

Figure 15: DBPedia. All models demonstrate a negative trend between distance and iteration. The figure for SciPhi displays a plateau between the second and third iteration, which could imply that the probing set (i.e. the actual text) or the semantics did not change much.

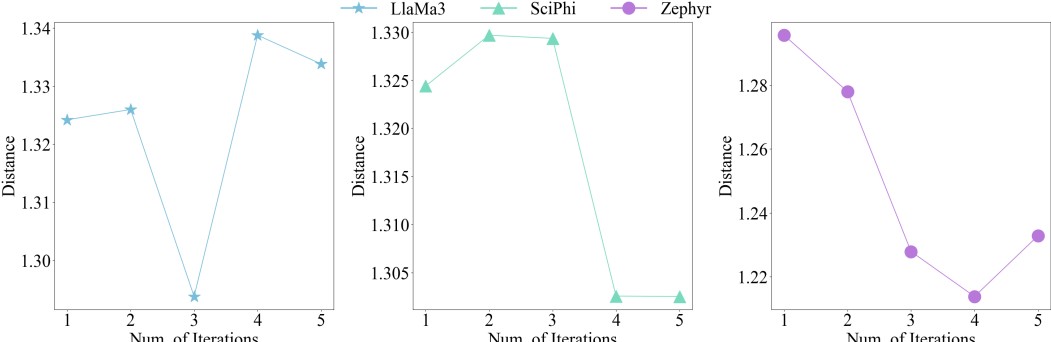

Figure 16: MPQA. The figures in general demonstrate a negative trend. For SciPhi, the distance increases first then drops after the second iteration. However, the difference is relatively small, about 0.05 difference, indicating that the generated samples are similar to the demonstrations.

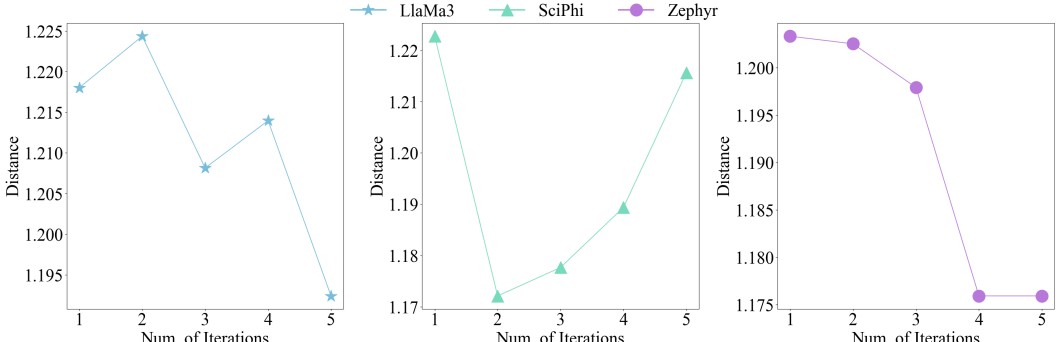

Figure 17: MR. For LlaMa3, the distance peaks at iteration two and iteration four, but generally decreases. This could be due to HIDO trying to find the best order in the neighborhood space but selecting one that does not perform well; however, it is able to find the best order in the end.

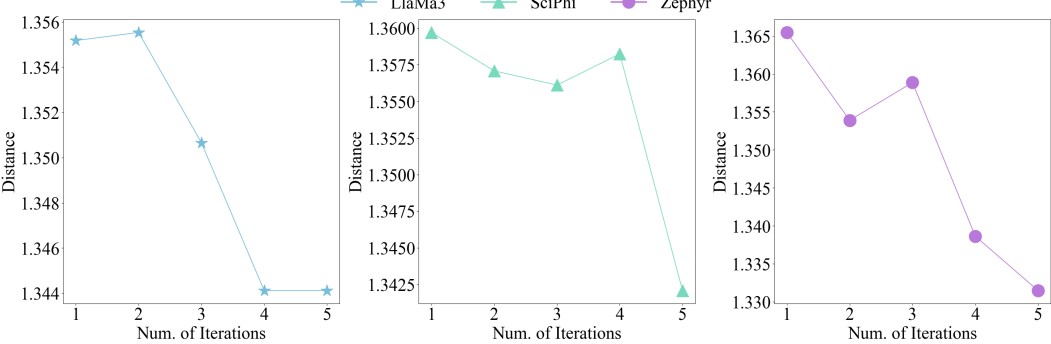

Figure 18: TREC. Like before, the genearl trend is negative in all the figures. However, the plots for SciPhi and Zephy both have a peak but drops in the next iteration, which indicates that the model diverges from the optimial and corrects itself.

