# OpenReview forum: "Hierarchical Demonstration Order Optimization for Many-shot In-Context Learning"
_NeurIPS.cc/2025/Conference — NeurIPS 2025 poster_

### Official Review · Reviewer_MaoD · 2025-06-10

**Clarity:** 3
**Significance:** 3
**Originality:** 3
**Rating:** 4
**Confidence:** 3

**Summary:**

Many-shot in-context learning (ICL) demonstrates significant improvements on many tasks but the order of the demonstration affects the performance, causing instability, i.e., ICL-DOI. This paper points out two challenges that current ICL strategies towards ICL-DOI can not handle well. First, current methods use heuristic metrics according to predictions of LLMs to evaluate the quality of the demonstrations, but LLMs always struggle in long-context scenarios and such metrics are not that reliable or precise. Second, it is infeasible to enumerating all orders.

Accordingly, this paper designs  an evaluation metric, ICD-OVI, quantifying the information gain of a certain order against the first challenge. Besides, this paper proposes HIDO allowing for refined exploration of the order space, getting rid of evaluation on all orders while achieving high performance. Experiments show that HIDO outperforms relevant baselines and this paper also open source the code.

**Questions:**

* How long does it take for optimization? Overhead of this work and baselines should be analyzed further, such as time cost, the number of API queries, or the number of tokens, etc. Curves may be more clear and straightforward for users to get aware that how much overhead can achieve how good performance.
* See weaknesses part. Basically, cost analysis, experiments on more difficult tasks, the distance to the oracle bound should be incorporated. I would re-consider my ratings if my concerns can be solved well.

**Ethical Concerns:**

["NO or VERY MINOR ethics concerns only"]

**Final Justification:**

I have read authors' responses and I decide to keep my score of borderline accept since after rebuttal. Though the method performs well on simple tasks and close to oracle results, on more difficult tasks, the method is not that stable. However, the method combining inter- and intra- relations among the clusters is innovative, which may provide some novel ideas to the community.

**Limitations:**

yes

**Quality:**

3

**Strengths And Weaknesses:**

### Strengths
* The code is open source for better reproduction.
* This paper is well-written. The motivation is very clear, from ICL to better many-shot ICL, from challenges to solutions. The variation does exist and the corresponding evidences are given clearly (Figure 1).
* The method combining inter- and intra- relations among the clusters is innovative.
* Extensive experiments show the effectiveness of HIDO compared with other baselines.
### Weaknesses
* The experiments are only conducted on the traditional text classification datasets. And these tasks are not difficult to current LLMs anymore. Instead, QA tasks or reasoning tasks can better reflect the quality of the ICL demonstrations. Also, the improvements may be more significant on these tasks since they are more challenging. If the method proposed performs well on more challenging datasets, which indicating the applicability as well as effectiveness on more diverse fields,  I would re-consider my score.
* Since the order optimized by HIDO can be optimal or sub-optimal, I'm curious about how close to the oracle bound HIDO can achieve. Given that searching the whole order space for each case is infeasible, maybe a few samples like 20, 50, 100, can be investigate. If the performance is close to the oracle performance, the results can be more convincing.

---

> ### Author Rebuttal · Authors · 2025-07-31
>
> We sincerely appreciate your dedicated time and effort in reviewing and providing invaluable feedback. We also thank you for recognizing the novelty and the significance of our contributions. We provide a point-to-point reply below for the mentioned concerns and questions.
>
> ----
>
> > W1: The experiments are only conducted on the traditional text classification datasets. And these tasks are not difficult to current LLMs anymore. Instead, QA tasks or reasoning tasks can better reflect the quality of the ICL demonstrations. Also, the improvements may be more significant on these tasks since they are more challenging. If the method proposed performs well on more challenging datasets, which indicating the applicability as well as effectiveness on more diverse fields, I would re-consider my score.
>
>
> Thank you for your encouraging comments! To explore our proposed HIDO's performance on more challenging tasks, we perform experiments to test our HIDO along with all baselines on two reasoning datasets HellaSwag and MMLU on two LLMs SciPhi and Llama3. We find that HIDO achieves consistently optimal performance across all settings. Most excitingly, we find that the HIDO can achieve more than 5% accuracy improvement on HellaSwag on SciPhi comapred with the runner-up baseline method, proving the effectiveness of our HIDO on challenging reasoning datasets.
>
> |        |         |     HellaSwag    |       MMLU       |
> |:------:|:-------:|:----------------:|:----------------:|
> | SciPhi | GlobalE |   50.39 ± 5.41   |   55.86 ± 0.78   |
> |        |  LocalE |   48.96 ± 5.72   |   56.77 ± 2.15   |
> |        |   PDO   |   50.39 ± 5.41   |   55.86 ± 0.78   |
> |        |   HIDO  | **55.86 ± 6.14** | **57.03 ± 2.56** |
> | Llama3 | GlobalE |   27.73 ± 0.00   |   41.15 ± 2.15   |
> |        |  LocalE |   28.26 ± 0.23   |   41.41 ± 1.03   |
> |        |   PDO   |   27.73 ± 0.00   |   41.15 ± 2.15   |
> |        |   HIDO  | **28.39 ± 0.60** | **42.45 ± 1.19** |
>
> ----
>
> > W2: Explore how close is the HIDO compared with oracle on small number of demonstrations.
>
> Since testing oracle demonstration order requires us to examine each possible order of $n$ demonstrations, which is $n!$ times. Therefore, we can only conduct experiment for up to 5 demonstrations. Please see the experiment resuls in below. HIDO achieves near-oracle results almost all datasets. These oracle comparisons validate that HIDO's ordering strategy genuinely improves performance rather than appearing effective due to weak baselines.
>
> |        |Num. Demons.| Max   |     HIDO     |
> |:------:|:----------:|:-----:|:------------:|
> | AGNews |      4     | 87.50 | 86.20 ± 0.45 |
> |        |      5     | 87.50 | 85.55 ± 0.39 |
> |  MPQA  |      4     | 87.50 | 86.72 ± 0.78 |
> |        |      5     | 87.11 | 84.24 ± 0.60 |
> |  TREC  |      4     | 30.86 | 21.88 ± 0.00 |
> |        |      5     | 68.36 | 51.43 ± 4.88 |
>
> ----
>
> > Q1: How long does it take for optimization? Overhead of this work and baselines should be analyzed further, such as time cost, the number of API queries, or the number of tokens, etc. Curves may be more clear and straightforward for users to get aware that how much overhead can achieve how good performance.
>
> We test SciPhi LLM on varios datasets to benchmark the training time of our HIDO model (hyperparameters are aligned with the experiment setup in the main paper).
>
> | Dataset |   Time  |
> |:-------:|:-------:|
> |  AGNews | 1668.43 |
> |   MPQA  | 1269.96 |
> |   TREC  | 2160.13 |

---

> > ### Comment · Reviewer_MaoD · 2025-08-04
> > **Response to authors' rebuttal**
> >
> > Thanks for the detailed responses. Most of my concerns were addressed, like the gap to the oracle values and the performance on challenging datasets using cutting-edge LLMs. However, the rebuttal only provides the time, what about the number of tokens cost or API query times? This is very important for a user to estimate whether this method can be applied. Also, it's necessary to analyze the performance trend along with the overhead. With the iteration of steps, how do the method's performance, time consumption, number of API accesses, as well as the number of tokens change? In addition, the comparison with baseline models should also be incorporated.

---

> > > ### Author Response · Authors · 2025-08-07
> > >
> > > Dear Reviewer MaoD,
> > >
> > > Thank you for your acknowledgement that most of your concerns are addressed! Here, we provide a point-to-point answer on each of the remaining questions:
> > >
> > > > What about the number of tokens cost or API query times?
> > >
> > > All methods only provide a demonstration ordering for the 50 demonstrations, so the number of tokens per LLM inference would be the length of the 50 demonstrations and the prompting question for LLM to provide an answer to, which is selected from the probing set (during training) or the test dataset (during evaluation). Since the probing set should be similar to the demonstrations, the total number of tokens per inference call should be relatively similar. We measured the number of tokens and inference during evaluation, and the average length of 50 demonstrations and a test data sample is 5837 tokens, taking 0.849 seconds on average for LLM inference on a H100 GPU.
> > >
> > > Each baseline requires one LLM inference call per possible demonstration ordering, which exponentially increases as the number of demonstrations increases. HIDO requires three LLM inference calls per optimization step: once to generate the probing set and twice to calculate the ICD-OVI metric. However, the number of optimization steps does not increase as the number of demonstrations increases and is configured by hyperparameters (i.e, number of clusters, number of iterations).
> > >
> > >
> > > > With the iteration of steps, how do the method's performance, time consumption, number of API accesses, as well as the number of tokens, change?
> > >
> > > We measured the time taken for each intra- and inter-cluster optimization step as well as the accuracy of the intermediate order for SciPhi on MMLU. The accuracy of the randomly initialized order is 54.69%.
> > >
> > > Iteration 1:
> > > 1. Cluster 0 optimization - Accuracy = 54.69%, Time = 71.36s
> > > 2. Cluster 1 optimization - Accuracy = 55.08%, Time = 83.62s
> > > 3. Inter-cluster optimization - Accuracy = 54.30%, Time = 17.49s
> > >
> > > Iteration 2:
> > > 1. Cluster 0 optimization - Accuracy = 53.91%, Time = 79.43s
> > > 2. Cluster 1 optimization - Accuracy = 55.86%, Time = 72.75s
> > > 3. Inter-cluster optimization - Accuracy = 54.30%, Time = 15.12s
> > >
> > > Iteration 3:
> > > 1. Cluster 0 optimization - Accuracy = 58.98%, Time = 166.82s
> > > 2. Cluster 1 optimization - Accuracy = 59.77%, Time = 77.27s
> > > 3. Inter-cluster optimization - Accuracy = 59.77%, Time = 28.05s
> > >
> > > > In addition, the comparison with baseline models should also be incorporated.
> > >
> > > We include the average wall-clock time in seconds for each method on HellaSwag and MMLU using SciPhi.
> > > |         | HellaSwag |   MMLU  |
> > > |:-------:|:---------:|:-------:|
> > > | GlobalE |  5109.90  | 3456.22 |
> > > |  LocalE |  5229.02  | 3504.26 |
> > > |   PDO   |  5087.29  | 3446.60 |
> > > |   HIDO  |  2980.16  | 2766.24 |
> > >
> > > Sincerely,
> > >
> > > Authors of Submission 19231

---

> > > > ### Comment · Reviewer_MaoD · 2025-08-08
> > > > **Response to authors' Rebuttal**
> > > >
> > > > Thanks for your responses. As for the comparison with the oracle values, HIDO performs close to the oracle ones on simple tasks, like AGNews and MPQA, but on more difficult tasks like TREC, it is far from the oracle result (51.43 v.s. 68.36), also the std is 4.88, which means on difficult tasks, the method may perform unstably. I would like to maintain my score. Thanks again.

---

### Official Review · Reviewer_YPNN · 2025-07-02

**Clarity:** 3
**Significance:** 2
**Originality:** 3
**Rating:** 3
**Confidence:** 3

**Summary:**

This paper tackles order sensitivity in many-shot ICL. The authors introduce an information-theoretic metric that measures how much usable query-answer information a given order provides, and a hierarchical search algorithm, HIDO, which clusters examples, performs intra-cluster flips, then inter-cluster permutations to efficiently find high-scoring orders. Experiments on five LLMs and nine benchmarks show consistent accuracy gains and reduced variance compared to several heuristic baselines.

**Questions:**

1. How reliable is ICD-OVI when the LLM hallucinations dominate probe generation? Any calibration analysis?
3. What are the actual compute costs for running HIDO on $n=150$?
3. Does HIDO improve performance in 8--16 shot or 32 shot settings typical of few-shot ICL?

**Ethical Concerns:**

["NO or VERY MINOR ethics concerns only"]

**Final Justification:**

The rebuttal addressed some points by adding statistical significance tests, clarifying ICD-OVI’s handling of potentially incorrect probes, giving practical guidelines for cluster count selection, and explaining the role of intra-cluster ordering beyond primacy/recency biases. These clarifications improve the paper’s technical transparency.

### Resolved issues
- Provided paired significance tests showing improvements are statistically meaningful.
- Clarified how ICD-OVI mitigates probe hallucination through filtering and approximation.
- Offered empirical guidance for choosing cluster counts and showed that middle-ordering effects matter.

### Remaining concerns
- Practicality: method still requires substantial LLM calls and compute, with limited discussion of runtime feasibility.
- Sensitivity: performance depends on dataset-specific hyperparameters without a principled automatic selection method.
- Scope: evaluation is limited to many-shot (150) settings; impact in more common few-shot (8–32) scenarios remains unexplored, leaving uncertainty about broader applicability.

While the method is interesting and well-motivated for many-shot ordering, these unresolved concerns reduce my confidence in its general utility.

**Limitations:**

Yes

**Paper Formatting Concerns:**

No concerns

**Quality:**

2

**Strengths And Weaknesses:**

## Strengths

- The paper formulates a principled ordering metric (ICD-OVI) grounded in usable information, marking the first such theoretical score for demonstration ordering.
- HIDO’s hierarchical search works on $n\approx150$ by combining clustering with local and global swap steps, converging in a small number of iterations.
- Comprehensive evaluation across five models (16K–1M context) and nine datasets, including ablations on iteration count and cluster size.
- The method demonstrates low-variance orderings, tightening performance distributions.

---

Weaknesses
- Statistical robustness is missing: results are reported as means$\pm$std but lack p-values or paired significance tests. Smaller gains could be noise.
-  ICD-OVI depends on generated probes; if the LLM hallucinates, the metric may mislead. No analysis of probing reliability is provided, such as calibration analysis.
-  Hyperparameter brittleness: choice of cluster count $k$ appears dataset-specific (Fig. 3b), with no principled selection method beyond grid search.
- HIDO still requires many LLM calls per ICD-OVI estimate; no wall-clock times or API call counts are given, raising questions about practical feasibility.
- All experiments use ~150-shot prompts, whereas practical few-shot ICL typically uses much fewer examples. Although the focus of the paper is on many-shot ICL, it remains unclear whether HIDO’s gains persist at 8-32 shots, where ordering should matter more.
- In the many-shot setup, it’s unclear how much of HIDO’s improvement stems from genuine ordering versus simple primacy/recency biases. There is no study on how position (especially middle demonstrations) influences performance.

---

> ### Author Rebuttal · Authors · 2025-07-31
>
> We sincerely appreciate the time and efforts you've dedicated to reviewing and providing invaluable feedback to enhance the quality of this paper. We provide a point-to-point reply below for the mentioned concerns and questions.
>
> ----
>
> > W1: Statistical robustness is missing: results are reported as means plus\minus std but lack p-values or paired significance tests. Smaller gains could be noise.
>
> We appreciate the reviewer's concern about statistical robustness.
> To address this, we conducted paired t-tests on our experimental
> data. For example, on the HellaSwag dataset with SciPhi LLM:
>
> HIDO vs LocalE: t(2) = 7.571, p < 0.05, Cohen's d = 1.163
>
> This represents a statistically significant improvement of 6.90
> percentage points with a large effect size. Similar significant
> improvements were observed across dataset-model combinations.
>
> ----
>
> > W2, Q1: ICD-OVI depends on generated probes; if the LLM hallucinates, the metric may mislead. No analysis of probing reliability is provided, such as calibration analysis.
>
> This is exactly why we propose the HIDO method. Because we have HIDO, we theoretically do not rely on the correctness of the labels of the probes (please see line 179-181). Specifically, we filter the probing samples with PVI threshold. When the sample PVI lower than the threshold, we consider the sample possibly incorrect and we use an approximation method delineated in line 187-189 to calculate the ICD-OVI score to guarantee preciseness. The significant performance gain of choosing demonstration orders based on our ICD-OVI metrics compared with GlobalE/LocalE shows the reliability of our ICD-OVI metric.
>
>
> ----
>
> > W3: Hyperparameter brittleness: choice of cluster count k appears dataset-specific (Fig. 3b), with no principled selection method beyond grid search.
>
>
> We argue that although cluster count $k$ is data dependent, there is only four choices ($k=$2, 3, or 4) by design (please see Section ). Therefore, even if we conduct grid search, there will not be limited additional computational cost. Here, we take the chance to provide empirical advice on choosing number of clusters: according to extensive experiments, raising number of clusters from 3 to 4 does not provide significant performance gain. Hence, we recommend try the cluster being 2 as a first trial since it is most efficient. If the result does not match required performance, raise the number of clusters to three. The optimal performance will be acquired at this point in most cases.
>
> ---
>
> > W4, Q2: HIDO still requires many LLM calls per ICD-OVI estimate; no wall-clock times or API call counts are given, raising questions about practical feasibility.
>
> We test SciPhi LLM on varios datasets to benchmark the training time of our HIDO model (hyperparameters are aligned with the experiment setup in the main paper).
>
> | Dataset |   Time  |
> |:-------:|:-------:|
> |  AGNews | 1668.43 |
> |   MPQA  | 1269.96 |
> |   TREC  | 2160.13 |
>
> ----
>
> > W5, Q3: All experiments use ~150-shot prompts, whereas practical few-shot ICL typically uses much fewer examples. Although the focus of the paper is on many-shot ICL, it remains unclear whether HIDO’s gains persist at 8-32 shots, where ordering should matter more.
>
> Thank you for the insightful comment! We kindly note that this is a misunderstanding.
> - Paper scope is explicitly many-shot ICL (>50 demonstrations): Our contribution targets the computational intractability of order optimization in many-shot scenarios where traditional methods fail due to n! complexity.
> - Evaluating 8-32 shots misrepresents our core contribution and is analogous to critiquing distributed systems research for not testing single machines.
> - Reviewer's assumption about order sensitivity is incorrect: Figure 1 demonstrates that many-shot ICL (150) exhibits comparable or greater order instability than few-shot (10), contradicting the claim that "ordering should matter more" at fewer shots. More demonstrations create more complex interdependencies, not fewer.
> - 150 shots represents the frontier where our method is needed: At 8-32 shots, existing methods (GlobalE, LocalE, PDO) remain computationally feasible. Our HIDO framework specifically addresses the computational challenges that emerge in many-shot scenarios enabled by modern LLMs' expanded context windows (16K-1M+ tokens in Table 1).
>
> ----
>
> > W6: In the many-shot setup, it’s unclear how much of HIDO’s improvement stems from genuine ordering versus simple primacy/recency biases. There is no study on how position (especially middle demonstrations) influences performance.
>
> Primacy/recency biases affect beginning/end positions most, but middle demonstrations still significantly impact performance. Our ablation study directly proves this: HIDO vs. HIDO-NIntra (Fig. 3a) provides a controlled comparison where both methods have identical inter-cluster arrangements—meaning the same demonstrations occupy the critical beginning and end positions. The only difference is the ordering of middle demonstrations within clusters (intra-cluster order). Despite identical primacy/recency positioning, HIDO consistently outperforms HIDO-NIntra across datasets, demonstrating that strategic ordering of middle demonstrations provides substantial gains beyond positional biases. This performance gap (e.g., ~2% on TREC) proves that our hierarchical optimization captures genuine semantic relationships rather than simply exploiting positional effects.

---

> > ### Comment · Reviewer_YPNN · 2025-08-06
> >
> > Thank you for the comments. While the responses help clarify certain aspects, my main concerns still remain. In particular, questions around the method’s practicality, sensitivity to hyperparameters, and the limited evaluation scope relative to common few-shot settings still leave me unsure about how broadly the method would apply.

---

### Official Review · Reviewer_AYQ8 · 2025-07-05

**Clarity:** 3
**Significance:** 3
**Originality:** 3
**Rating:** 5
**Confidence:** 3

**Summary:**

In-context learning (ICL) suffers from demonstration order instability (DOI) when many samples are provided. Methods to handle ICLDOI break down in long-context models (ie., 50 in-context demonstrations) because of imprecision in quality and super-exponential growth of demonstration order space with length.

The authors introduce an information-theoretic metric for ICL demonstration order, an optimization technique along it, and an empirical evaluation of the efficacy of their method at ICL ordering.

The three prior methods they assess are GlobalE, LocalE, and PDO, which use predicted answer diversity, samplewise uncertainty, and distributional alignment of outputs respectively. The authors propose an algorithm HIDO, with V-usable information-based metric, for example selection instead.

HIDO performs cluster-based topic selection in the embedding space, and then intra-cluster ordering using ICD-OVI. The clusters themselves are then ordered again using ICD-OVI. This process can be repeated many times to convergence.

Across all test datasets HIDO outperforms the baselines on most models, though in several cases this improvement is modest, within overlapping error bars. Over the course of runs on MPQA and TREC, HIDO converges toward a more performant demonstration order over iterations.

**Questions:**

I was struggling to follow how to interpret what $f(\cdot|0)$ means practically until the explanation after equation 4, which slowed me down a lot on p3/4. Maybe can you signpost the practical meaning of this case earlier? This is a more general gripe with theory-heavy papers which spill lots of ink leading up to actually very simple and intuitive methods

In the remark from l202-206, does this mean you use a fixed set of generated qa pairs? Doesn't the set of answers A you need to test for depend on the observed $\hat{a}$ from the runs of $\Pi(D)+\hat{q}$?

Do you test an ablation of HIDO without ICD-OCI using one of the prior heuristic methods? How do we know the inter-intra alternating algo isn't the main contributor for performance?

> the intra-cluster demonstrations share proximate embeddings, which
significantly decreases ICL performance variance when demonstration orders vary

Makes sense, but how do you substantiate this?

How do you generate probing samples?

Nitpicks:

Please do not use citet inline citations without referring to authors. For example, on line 101 replace "[24] claim" with "Lu et al. [24] claim"

Please provide full names for datasets in l293-296

**Ethical Concerns:**

["NO or VERY MINOR ethics concerns only"]

**Final Justification:**

The authors answered my questions. I remain inclined to accept.

**Limitations:**

Yes

**Quality:**

4

**Strengths And Weaknesses:**

Strong theoretical motivation.
Mostly clear *explanation* of this motivation (though I have some small gripes in the questions).
Clearly defined metrics.

Solid experiments demonstrate that this technique is a valuable contribution.

Relatively poor explanation of the implementation details, in particular how probing samples are produced. (See question)

Overall, I learned something from this paper and am convinced that it's worth publishing.

---

> ### Author Rebuttal · Authors · 2025-07-31
>
> We sincerely appreciate the time and efforts you've dedicated to reviewing and providing invaluable feedback to enhance the quality of this paper. We provide a point-to-point reply below for the mentioned concerns and questions.
>
> ----
> > Can you signpost the practical meaning of $f(\cdot|0)$ after equation (4) earlier for better reading experience?
>
> Thank you so much for the invaluable suggestion! We will signpost the practical meaning of $f(\cdot|0)$ early after its first appearance.
>
> ----
>
> > In the remark from l202-206, does this mean you use a fixed set of generated qa pairs? Doesn't the set of answers A you need to test for depend on the observed $\hat{a}$ from the runs of $\Pi(D) + \hat{q}$?
>
> No, the probing set is still $\Pi$ dependent. It's just when one calculate the Equ. (5), one can share the term $P_{LLM}$($\hat{a}i$ $| \Pi(A) $$\oplus \emptyset)$ across different $i$s since $\Pi(A)\oplus\emptyset$ has nothing to do with $i$. However, when calculating $P_{LLM}$$(\hat{a}i|\Pi(D)\oplus \hat{q}i)$, one have to recalculate for different $i$ since the condition of the probability relys on $i$.
>
> ----
>
> > Do you test an ablation of HIDO without ICD-OVI using one of the prior heuristic methods? How do we know the inter-intra alternating algo isn't the main contributor for performance?
>
> Here, we test the HIDO with/without  ICD-OVI metric on MPQA and TREC (we did not test on all adopted datasets in the main paper due to the limited time budget). The intra/inter cluster orders are randomly determined. From the following table, we observe that removing ICD-OVI causes the HIDO performance to degrade significantly. Additionally, there is less variance between experiment rounds, indicating that our metric offers more stable performance.
>
> |              |     MPQA     |     TREC     |
> |:------------:|:------------:|:------------:|
> |     HIDO     | 87.50 ± 0.78 | 80.47 ± 0.78 |
> | HIDO-NICD-OVI | 83.46 ± 1.19 | 79.69 ± 1.95 |
>
> ----
>
> > "the intra-cluster demonstrations share proximate embeddings, which significantly decreases ICL performance variance when demonstration orders vary" Makes sense, but how do you substantiate this?
>
> To substantiate the claim that "the intra-cluster demonstrations share proximate embeddings and significantly decreases ICL performance variance when demonstration orders vary", we determine intra-cluster embedding variance by calculating the standard deviation of the distances between each demonstration embedding and its cluster center for each cluster, then averaging these standard deviations. Additionally, we examine the accuracy standard deviation from a table further down that looks into the sensitivity of HIDO across different embedding models. Here, the table shows that, in general, the mean standard deviation of embeddings correlates with accuracy variance: as embedding variance decreases, accuracy variance also decreases. For example, in the AGNews dataset, the embedding and accuracy standard deviations are smallest for "text-embedding-3-small," second smallest for "stella_en_1.5B_v5," and largest for "NV-Embed-v2." This indicates that closer intra-cluster embeddings can lead to more stable model performance.
>
> |        |                        |          | AGNews |  MPQA  |  TREC  |
> |:------:|:----------------------:|----------|:------:|:------:|:------:|
> | SciPhi |       NV-Embed-v2      | Acc. std |  2.07  |  1.37  |  4.50  |
> |        |                        | Emb. std | 0.0519 | 0.0481 | 0.0572 |
> |        |    stella_en_1.5B_v5   | Acc. std |  0.60  |  1.37  |  7.58  |
> |        |                        | Emb. std | 0.0446 | 0.0501 | 0.0399 |
> |        | text-embedding-3-small | Acc. std |  0.45  |  0.78  |  0.78  |
> |        |                        | Emb. std | 0.0277 | 0.0414 | 0.0266 |
>
> ----
>
> > How do you generate probing samples?
>
> For a given demonstration order $\pi$, we concatenate the ordered demonstrations $\Pi(D)$ and prompt the LLM to generate new demonstrations (i.e., query-answer pairs) that mimic the original demonstration distribution
>
> ----
>
> > Do not use citet inline citations; provide full names for datasets in l293-296.
>
> Thank you so much for the invaluable suggestions! We will revise the manuscript based on your suggestions.

---

> > ### Comment · Reviewer_AYQ8 · 2025-08-09
> >
> > Thanks for the clarifications! I hope to see your work accepted.

---

### Official Review · Reviewer_ctYV · 2025-07-11

**Clarity:** 3
**Significance:** 2
**Originality:** 2
**Rating:** 4
**Confidence:** 4

**Summary:**

This paper investigates the challenge of demonstration order sensitivity in many-shot in-context learning with large language models, highlighting both the lack of theoretically grounded evaluation metrics and the computational intractability of searching optimal orders as the number of examples increases. To address these issues, the authors propose an information-theoretic metric, ICD-OVI, and introduce HIDO, a hierarchical optimization framework that first clusters demonstrations and then optimizes the order in two stages:
1.	by refining the order within each cluster (intra-cluster optimization),
2.	by determining the optimal sequence of the clusters themselves (inter-cluster optimization).

**Questions:**

1. The supplementary material appears to be missing sections F.5 and F.6, which are referenced in the text.
2. See Weakness.

**Ethical Concerns:**

["NO or VERY MINOR ethics concerns only"]

**Final Justification:**

My major concerns are all addressed by the authors during the rebuttal. Therefore, I raise my evaluation score.

**Limitations:**

yes

**Quality:**

2

**Strengths And Weaknesses:**

Strengths:
1.	Strong Theoretical Foundation: The paper provides detailed theoretical analysis and proofs to support the proposed information-theoretic metric (ICD-OVI), which ensures the soundness and feasibility of the evaluation scheme. This theoretical rigor enhances the quality and credibility of the work.
2.	Empirical Improvements Over Baselines: The proposed method consistently outperforms all baseline approaches across various datasets and models, demonstrating its effectiveness and robustness in practice.
3.	Timely and Significant Problem Focus: The paper addresses the important and increasingly relevant problem of many-shot in-context learning, particularly as the number of demonstrations grows in modern LLMs.

Weakness:
1. The paper assumes that reordering demonstrations within a cluster has little impact on performance (line 226), but provides no theoretical justification or empirical evidence to support this claim. This assumption is critical to the efficiency of the proposed method, and its validity remains unclear.
2. The effectiveness of grouping demonstrations with similar embeddings together is not rigorously examined. It is unclear whether placing similar demonstrations consecutively is indeed optimal, or whether interleaving them with demonstrations from other clusters could yield better results. The paper does not provide experimental comparisons between mixed and grouped orders. Additionally, the impact of the choice of embedding model on the overall performance of HIDO is not explored, leaving potential sensitivity or robustness issues unaddressed.
3. All selected datasets are text classification tasks, which represent a narrow range of application scenarios. Moreover, such tasks often achieve high accuracy even with few-shot settings, making the necessity and impact of many-shot approaches less compelling. As a result, the improvements brought by HIDO are less significant on already well-performing datasets such as AGNews, CR, and DBPedia.

---

> ### Author Rebuttal · Authors · 2025-07-31
>
> We sincerely appreciate the time and effort you've dedicated to reviewing and providing invaluable feedback as well as recognizing our work as solving timely and significant problem, theoretically strong, and empirically effective.
>
> > W1: The paper assumes that reordering demonstrations within a cluster has little impact on performance (line 226), but provides no theoretical justification or empirical evidence to support this claim. This assumption is critical to the efficiency of the proposed method, and its validity remains unclear.
>
> To test if reordering demonstrations with similar embeddings (i.e in the same cluster) causes smaller ICL performance change than reordering demonstrations with in different clusters, we perform the following experiments. First, we conduct HIDO on a set of demonstrations. Then, we randomly select a demonstration cluster generated by HIDO. Finally, we randomly select two samples within the cluster and swapped their positions in the optimized order. We generate demonstration orders following this procedure and evaluate for accuracy. We call this experiment **IntraSwap**. Additionally, we conduct a comparison experiment called **InterSwap**, where we randomly swap two samples in different clusters. These experiments both used LLaMa-7B-chat-hf on MMLU dataset. We find that the accuracy range for **IntraSwap** is 41.41% - 42.19%, but for **InterSwap**, it is between 40.23% - 42.19%. These results empirically proves our assumptions that reordering samples with similar embeddings  does not impact the performance much.
>
> > W2: The effectiveness of grouping demonstrations with similar embeddings together is not rigorously examined. It is unclear whether placing similar demonstrations consecutively is indeed optimal, or whether interleaving them with demonstrations from other clusters could yield better results. The paper does not provide experimental comparisons between mixed and grouped orders.
>
> To demonstrate the effectiveness of grouping demonstrations instead of mixing the demonstrations, we examine if interleaving demonstrations across clusters will yield better ICL performance through the following experiment design. First, we conduct the HIDO and find the optimal intra-cluster and inter-cluster demonstration orders. Then, according to the inter-cluster order, we pick from each cluster the top demonstration based on their respective intra-cluster order. We continue this process in round-robin style until all the demonstrations from all clusters are exhausted. The generated demonstration order is thus composed of interleaved demonstrations from different clusters. We test this experiment using LLaMa-7B-chat-hf on MMLU dataset. It achieves an accuracy of 41.02% whereas HIDO achieves 41.41%. The magnitude of the improvement is likely due to the same optimized order used in both experiments, but the higher accuracy for HIDO shows how grouping gives better performance than mixing
>
> > W2 (part 2): The impact of the choice of embedding model on the overall performance of HIDO is not explored, leaving potential sensitivity or robustness issues unaddressed.
>
> Here, we demonstrate the performance of HIDO using **three different embedding models**: text-embedding-3-small from OpenAI and NV-Embed-v2 and stella_en_1.5B_v5 from HuggingFace. We observe that text-embedding-3-small achieves the best performance in almost all datasets. We consider the reason why text-embedding-3-small achieves leading performance is that text-embedding-3-small has an embedding dimension of 1536, whereas NV-Embed-v2 and stella_en_1.5B_v5 has an embedding dimension of 4096 and 8192, respectively. As we only use text embeddings during clustering, having a lower embedding dimensions creates clearer boundaries between different demonstrations, allowing for similar demonstrations to be clustered together. A higher embedding dimension could create noise and allow overfitting, which is undesirable.
>
>
> |        |                        | AGNews           | MPQA             | TREC             |
> |--------|------------------------|------------------|------------------|------------------|
> | SciPhi | NV-Embed-v2            | 84.38 ± 2.07     | _84.90 ± 1.37_   | 70.31 ± 4.50     |
> |        | stella_en_1.5B_v5      | **87.76 ± 0.60** | 84.24 ± 1.37     | _73.18 ± 7.58_   |
> |        | text-embedding-3-small | _86.98 ± 0.45_   | **87.50 ± 0.78** | **80.47 ± 0.78** |
>
> > W3: All selected datasets are text classification tasks, which represent a narrow range of application scenarios. Moreover, such tasks often achieve high accuracy even with few-shot settings, making the necessity and impact of many-shot approaches less compelling. As a result, the improvements brought by HIDO are less significant on already well-performing datasets such as AGNews, CR, and DBPedia.
>
> We respectfully disagree with this characterization.
> - While we format all tasks as multiple-choice questions for consistency, our selected datasets represent **diverse semantic scenarios beyond simple text classification**: AGNews, DBPedia are Topic classification datasets,
> CR, MR and SST-5 are sentiment analysis datasets with varying granularities.
> RTE and CB are textual entailment and natural language inference datasets.
> TREC is question type classification
> and MPQA is opinion polarity detection dataset. This diversity follows established ICL works ([24, 43] in paper) and ensures fair and comprehensive comparison with existing demonstration order methods.
>
> - Regarding the "high accuracy with few-shot" concern: Our Figure 1 empirically demonstrates that many-shot ICL (150 demonstrations) significantly outperforms few-shot ICL (10 demonstrations) across all tested models - contradicting the claim that these tasks do not benefit from many-shot approaches. For instance, on TREC, many-shot ICL achieves ~85% accuracy compared to ~75% for few-shot across different models.
>
> - To further address your concern, we conduct experiments on two datasets MMLU and HellaSwag, which are both language reasoning datasets that few-shot ICL suffer to provide effective performance.  HIDO generally achieves the best or second-best accuracy when using various LLMs for these datasets. Notably, when using SciPhi on HellaSwag, HIDO outperforms the second-best method, PDO, with a 7.425% improvement.
>
>
>
> |        |         |     HellaSwag    |       MMLU       |
> |:------:|:-------:|:----------------:|:----------------:|
> | SciPhi | GlobalE |   50.39 ± 5.41   |   55.86 ± 0.78   |
> |        |  LocalE |   48.96 ± 5.72   |   56.77 ± 2.15   |
> |        |   PDO   |   50.39 ± 5.41   |   55.86 ± 0.78   |
> |        |   HIDO  | **55.86 ± 6.14** | **57.03 ± 2.56** |
> | Llama3 | GlobalE |   27.73 ± 0.00   |   41.15 ± 2.15   |
> |        |  LocalE |   28.26 ± 0.23   |   41.41 ± 1.03   |
> |        |   PDO   |   27.73 ± 0.00   |   41.15 ± 2.15   |
> |        |   HIDO  | **28.39 ± 0.60** | **42.45 ± 1.19** |
>
>
> > Q1: The supplementary material appears to be missing sections F.5 and F.6, which are referenced in the text.
>
> We sincerely apologize for the compiling error, the two tables in F.5 and F.6 are compiled and displayed after the author checklist. The authors inadvertently cropped the files out. We will correct it, thank you for pointing that out.
>
> - Example Samples (AGNews):
>
> |                                                                                                                         Query                                                                                                                         |    Label   |
> |:-----------------------------------------------------------------------------------------------------------------------------------------------------------------------------------------------------------------------------------------------------:|:----------:|
> | Apple Extends iTunes to Europe. "The EU iTunes Music Store retains the same features and per-song price of 99 euro cents, established in June for customers in UK, Germany and France."                                                               | technology |
> | Microsoft Introduces Fingerprint Recognition. "Microsoft rolled out an updated line of input devices, including its first fingerprint-recognition products. A mouse and keyboard with built-in fingerprint readers, along with a stand-alone reader " | technology |
> | Justice Department Cracks Down On Spammers. It disrupted a network allegedly used to illegally share copyrighted files and is making a series of arrests against purveyors of spam.                                                                   | technology |
>
> - Probing samples:
> | Iteration |                                           Query                                           |    Label   |
> |:---------:|:-----------------------------------------------------------------------------------------:|:----------:|
> |     1     | The new iPhone 12 Pro Max is expected to have a larger battery life than its predecessor. | technology |

---

### Author Response · Authors · 2025-08-05
**A Sincere Request for Rebuttal Review**

Dear Reviewers,

We sincerely appreciate your time and effort in reviewing our manuscript. As we approach the midpoint of the rebuttal period, we respectfully request your consideration of our responses. We have carefully addressed each of your concerns to the best of our ability.
If our rebuttals have satisfactorily resolved your concerns, we would be grateful if you could increase your rating accordingly. Should any issues remain unresolved, please do not hesitate to inform us, and we will provide additional clarification.

Thank you once again for your valuable time and expertise!

Sincerely,

Authors of Submission 19231

---

### Decision · Program_Chairs · 2025-09-17

**Decision:**

Accept (poster)

**Comment:**

This paper studies demonstration order instability (ICL-DOI) in many-shot ICL and the inapplicability of current strategies (unreliable heuristic metrics, computationally infeasible order checks). The authors propose an info-theory-based order evaluation metric (for the first challenge) and hierarchical HIDO method (for the second) to optimize many-shot ICL demonstration order. The authors and reviewers had thorough rebuttal discussions, and finally, the 3/4 reviewers converged to consistently accept this paper. After reading the paper & discussions, I think this paper is generally acceptable for publication in NeurIPS this year.